# Verification Based Solution for Structured MAB Problems

**Zohar Karnin**
Yahoo Research
New York, NY 10036
`zkarnin@ymail.com`

## Abstract

We consider the problem of finding the best arm in a stochastic Multi-armed Bandit (MAB) game and propose a general framework based on verification that applies to multiple well-motivated generalizations of the classic MAB problem. In these generalizations, additional structure is known in advance, causing the task of verifying the optimality of a candidate to be easier than discovering the best arm. Our results are focused on the scenario where the failure probability $\delta$ must be very low; we essentially show that in this high confidence regime, identifying the best arm is as easy as the task of verification. We demonstrate the effectiveness of our framework by applying it, and matching or improving the state-of-the art results in the problems of: Linear bandits, Dueling bandits with the Condorcet assumption, Copeland dueling bandits, Unimodal bandits and Graphical bandits.

## 1 Introduction

The Multi-Armed Bandit (MAB) game is one where in each round the player chooses an action, also referred to as an arm, from a pre-determined set. The player then gains a reward associated with the chosen arm and observes the reward while rewards associated with the other arms are not revealed. In the stochastic setting, each arm $x$ has a fixed associated value $\mu(x)$ throughout all rounds, and the reward associated with the arm is a random variable, independent of the history, with an expected value of $\mu(x)$. In this paper we focus on the pure exploration task [9] in the stochastic setting where our objective is to identify the arm maximizing $\mu(x)$ with sufficiently high probability, while minimizing the required number of rounds, otherwise known as the query complexity. This task, as opposed to the classic task of maximizing the sum of accumulated rewards is motivated by numerous scenarios where exploration (i.e. trying multiple options) is only possible in an initial testing phase, and not throughout the running time of the game.

As an example consider a company testing several variations of a (physical) product, and then once realizing the best one, moving to a production phase where the product is massively produced and shipped to numerous vendors. It is very natural to require that the identified option is the best one with very high probability, as a mistake can be very costly. Generally speaking, the vast majority of uses-cases of a pure exploration requires the error probability $\delta$ to be very small, so much so that even a logarithmic dependence over $\delta$ is non-negligible. Another example to demonstrate this is that of explore-then-exploit type algorithms. There are many examples of papers providing a solution to a regret based MAB problem where the first phase consists of identifying the best arm with probability at least $1 - 1/T$, and then using it in the remainder of the rounds. Here, $\delta = 1/T$ is often assumed to be the only non-constant.

We do not focus on the classic MAB problem but rather on several extensions of it for settings where we are given as input some underlying structural properties of the reward function $\mu$. We elaborate on the formal definitions and different scenarios in Section 2. Another extension we consider is that

of *Dueling Bandits* where, informally, we do not query a single arm but rather a pair, and rather than observing the reward of the arms we observe a hint as to the difference between their associated $\mu$ values. Each extension we discuss is motivated by different scenarios which we elaborate on in the upcoming sections. In all of the cases mentioned, we focus on the regime of high confidence meaning where the failure probability $\delta$ is very small.

Notice that due to the additional structure (that does not exist in the classic case), verifying a candidate arm is indeed the best arm can be a much easier task, at least conceptually, compared to that of discovering which arm is the best. This observation leads us to the following design: Explore the arms and obtain a candidate arm that is the best arm w.p. $1 - \kappa$ for some constant $\kappa$, then verify it is indeed the best with confidence $1 - \delta$. If the exploration procedure happened to be correct, the query complexity of the problem will be composed of a sum of two quantities. One is that of the exploration algorithm that is completely independent of $\delta$, and the other is dependent of $\delta$ but is the query complexity of the easier verification task. The query complexity is either dominated by that of the verification task, or by that of the original task with a *constant* failure probability. Either way, for small values of $\delta$ the savings are potentially huge. As it turns out, as discussed in Section 3, a careful combination of an exploration and verification algorithm can achieve an expected query complexity of $H_{\text{explore}} + H_{\text{verify}}$ where $H_{\text{explore}}$ is the exploration query complexity, independent of $\delta$, and $H_{\text{verify}}$ is the query complexity of the verification procedure with confidence $1 - \delta$. Below, we design exploration and verification algorithms for the problems of: Dueling bandits §4, Linear bandits §5, Unimodal graphical bandits §6 and Graphical bandits[1] . In the corresponding sections we provide short reviews of each MAB problem, and analyze their exploration and verification algorithms. Our results improve upon the state-of-the-art results in each of these mentioned problem (See Table 1 for a detailed comparison).

**Related Works:** We are aware of one attempt to capture multiple (stochastic) bandit problems in a single frameworks, given in [20]. The focus there is mostly on problems where the observed random variables do not necessarily reflect the reward, such as the dueling bandit problem, rather than methods to exploit structure between the arms. For example, in the case of the dueling bandit problem with the Condorcet assumption their algorithm does not take advantage of the structural properties and the corresponding query complexity is larger than that obtained here (see Section 4.1). We review the previous literature of each specific problem in the corresponding sections.

## 2 Formulation of Bandit Problems

The pure exploration Multi-Armed Bandit (MAB) problem, in the stochastic setting, can be generally formalized as follows. Our input consists of a set $\mathcal{K}$ of arms, where each arm $x$ is associated with some reward $\mu(x)$. In each round $t$ we play an arm $x_t$ and observe the outcome of a random variable whose expected value is $\mu(x_t)$. Other non-stochastic settings exist yet they are outside the scope of our paper; see [4] for a survey on bandit problems, including the stochastic and non-stochastic settings. The objective in the best arm identification problem is to identify the arm[2] $x^* = \arg\max \mu(x)$ while minimizing the expected number of queries to the reward values of the arms. Other than the classic MAB problem, where $\mathcal{K}$ is a finite set and $\mu$ is an arbitrary function there exist other frameworks where some structure is assumed regarding the behavior of $\mu$ over the arms of $\mathcal{K}$. An example for a common framework matching this formulation, that we will analyze in detail in Section 5, is that of the linear MAB. Here, $\mathcal{K}$ is a compact subset of $\mathbb{R}^d$, and the reward function $\mu$ is assumed to be linear. Unlike the classic MAB case, an algorithm can take advantage of the structure of $\mu$ and obtain a performance that is independent of the size of $\mathcal{K}$. Yet another example, discuss in Section 6, is that of unimodal bandits, where we are given a graph whose vertices are the arms, and it is guaranteed that the best arm is the unique arm having a maximal value among its neighbors in the graph.

The above general framework captures many variants of the MAB problem, yet does not capture the Dueling Multi Armed Bandit (DMAB) problem. Here, the input as before consists of a set of arms denoted by $\mathcal{K}$ yet we are not allowed to play a single arm in a round but rather a pair $x, y \in \mathcal{K}$. The general definition of the observation from playing the pair $x, y$ is a random variable whose

expected value is $P(x, y)$ where $P : \mathcal{K} \times \mathcal{K} \to \mathbb{R}$. The original motivating example for the DMAB [22] problem is that of information retrieval, where a query to a pair of arms is a presentation of the interleaved results of two ranking algorithms. The output is the 0 or 1, depending on the choice of the user, i.e. whether she chose a result from one or ranker or the other. The $\mu$ score here can be thought of a quality score for a ranker, defined according to the $P$ scores. We elaborate on the motivation for the MAB problem and the exact definition of the best arm in Section 4. In an extended version of this paper we discuss the problem of *graphical bandits* that is in some sense a generalization of the dueling bandit problem. There, we are not allowed to query any pair but rather pairs from some predefined set $E \subseteq \mathcal{K} \times \mathcal{K}$.

## 3 Boosting the Exploration Process with a Verification Policy

In what follows we present results for different variants of the MAB problem. We discuss two types of problems. The first is the well known pure exploration problem. Our input is the MAB instance, including the set of arms and possible structural information, and a confidence parameter $\kappa$. The objective is to find the best arm w.p. at least $1 - \kappa$ while using a minimal number of queries. We often discuss variants of the exploration problem where in addition to finding the best arm, we wish to obtain some additional information about the problem such as an estimate of the gaps of the reward value of suboptimal arms from the optimal one, the identity of important arm pairs, etc. We refer to this additional information as an advice vector $\theta$, and our objective is to minimize queries while obtaining a sufficiently accurate advice vector and the true optimal arm with probability at least $1 - \kappa$. For each MAB problem we describe an algorithm referred to as *FindBestArm* with a query complexity of[3] $H_{\text{explore}} \cdot \log(1/\kappa)$ that obtains an advice vector $\theta$ that is sufficiently accurate[4] w.p. at least $1 - \kappa$.

**Definition 1.** *Let* FindBestArm *be an algorithm that given the MAB problem and confidence parameter $\kappa > 0$ has the following guarantees. (1) with probability at least $1 - \kappa$ it outputs a correct best arm and advice vector $\theta$. (2) its expected query complexity is $H_{explore} \cdot \log(1/\kappa)$, where $H_{explore}$ is some instance specific complexity (that is not required to be known).*

The second type of problem is that of verification. Here we are given as input not only the MAB problem and confidence parameter $\delta$, but an advice vector $\theta$, including the identity of a candidate optimal arm.

**Definition 2.** *Let* VerifyBestArm *be an algorithm that given the MAB problem, confidence parameter $\delta > 0$ and an advice vector $\theta$ including a proposed identity of the best arm, has the following guarantees. (1) if the candidate optimal arm is not the actual optimal arm, the output is 'fail' w.p. at least $1 - \delta$. (2) if the advice vector is sufficiently accurate, and in particular the candidate is indeed the optimal arm, we should output 'success' w.p. at least $1 - \delta$. (3) if the advice vector is sufficiently accurate the expected query complexity is $H_{verify} \log(1/\delta)$. Otherwise, it is $H_{explore} \log(1/\delta)$.*

It is very common that $H_{\text{verify}} \ll H_{\text{explore}}$ as it is clearly an easier problem to simply verify the identity of the optimal arm rather than discover it. Our main result is thus somewhat surprising as it essentially shows that in the regime of high confidence, the best arm identification problem is as easy as verifying the identity of a candidate. Specifically we provide a complexity that is additive in $H_{\text{explore}}$ and $\log(1/\delta)$ rather than multiplicative. The formal result is as follows.

---

**Algorithm 1** Explore-Verify Framework

---

**Input:** Best arm identification problem, Oracle access to *FindBestArm* and *VerifyBestArm* with failure probability tuning, failure probability parameter $\delta$, parameter $\kappa$.
    **for all** $r = 1 \ldots$ **do**
        Call *FindBestArm* with failure probability $\kappa$, denote by $\theta$ its output.
        Call *VerifyBestArm* with advice vector $\theta$, that includes a candidate best arm $\hat{x}$, and failure probability $\delta/2r^2$. If succeeded, return $\hat{x}$. Else, continue to the next iteration
    **end for**

---

**Theorem 3.** *Assume that algorithm 1 is given oracle access to* FindBestArm *and* VerifyBestArm *with the above mentioned guarantees, and a confidence parameter $\delta < 1/3$. For any $\kappa < 1/3$, the algorithm identifies the best arm with probability $1 - \delta$ while using an expected number of at most*

$$O\left(H_{explore} \log(1/\kappa) + (H_{verify} + \kappa \cdot H_{explore}) \log(1/\delta)\right)$$

The following provides the guarantees for two suggested values of $\kappa$. The first may not be known to us but can very often be estimated beforehand. The second depends only on $\delta$ hence is always known in advance.

**Corollary 4.** *By setting $\kappa = \min\{1/3, H_{verify}/H_{explore}\}$, algorithm 1 has an expected number of at most*

$$O(H_{explore} \log(H_{explore}/H_{verify}) + H_{verify} \log(1/\delta))$$

*queries. By setting $\kappa = \min\{1/3, 1/\log(1/\delta)\}$, algorithm 1 has an expected query complexity of at most*

$$O(H_{explore} \log(\log(1/\delta)) + H_{verify} \log(1/\delta))$$

Notice that by setting $\kappa$ to $\min\{1/3, 1/\log(1/\delta)\}$, for any practical use-case, the dependence on $\delta$ in the left summand is nonexistent. In particular, this default value for $\kappa$ provides a multiplicative saving of either $H_{\text{explore}}/H_{\text{verify}}$, i.e. the ratio between the exploration and verification problem, or $\frac{\log(1/\delta)}{\log(\log(1/\delta))}$. Since $\log(1/\delta)$ is rarely a negligible term, and as we will see in what follows, neither is $H_{\text{explore}}/H_{\text{verify}}$, the savings are significant, hence the effectiveness of our result.

*Proof of Theorem 3.* In the analysis we often discuss the output of the sub-procedures in round $r > 1$, even if the algorithm terminated before round $r$. We note that these values are well-defined random variables regardless of the fact that we may not reach the round. To prove the correctness of the algorithm notice that since $\sum_{r=1}^{\infty} r^{-2} \leq 2$ we have with probability at least $1 - \delta$ that all runs of *VerifyBestArm* do not err. Since we halt only when *VerifyBestArm* outputs 'success' our algorithm indeed outputs the best arm w.p. at least $1 - \delta$

We proceed to analyze the expected query complexity, and start with a simple observation. Let $\text{QC}_{\text{single}}(r)$ denote the expected query complexity in round $r$, and let $Y_r$ be the indicator variable to whether the algorithm reached round $r$. Since $Y_r$ is independent of the procedures running in round $r$ and in particular of the number of queries required by them, we have that the total expected query complexity is

$$\mathbb{E}\left[\sum_{r=1}^{\infty} Y_r \text{QC}_{\text{single}}(r)\right] = \sum_{r=1}^{\infty} \mathbb{E}\left[Y_r\right] \cdot \mathbb{E}\left[\text{QC}_{\text{single}}(r)\right]$$

Hence, we proceed to analyze $\mathbb{E}\left[\text{QC}_{\text{single}}(r)\right]$ and $\mathbb{E}[Y_r]$ separately. For $\mathbb{E}\left[\text{QC}_{\text{single}}(r)\right]$ we have

$$\mathbb{E}\left[\text{QC}_{\text{single}}(r)\right] \leq H_{\text{explore}} \log(1/\kappa) +$$

$$\left((1-\kappa)H_{\text{verify}} + \kappa H_{\text{explore}}\right) \log\left(\frac{2r^2}{\delta}\right) \leq$$

$$H_{\text{explore}} \log(1/\kappa) + (\kappa H_{\text{explore}} + H_{\text{verify}}) \log\left(\frac{2r^2}{\delta}\right)$$

To explain the first inequality, the first summand is the complexity of *FindBestArm* . The second summand is that of *VerifyBestArm* , that is decomposed to the complexity in the scenario where *FindBestArm* succeeded vs. the scenario where it failed. To compute $\mathbb{E}[Y_r]$, we notice that $Y_r$ is an indicator function hence $\mathbb{E}[Y_r] = \Pr[Y_r = 1]$. In order for $Y_r$ to take the value of 1 we must have that for all rounds $r' < r$ either *VerifyBestArm* or *FindBestArm* have failed. Since the failure or success of the algorithms at different rounds are independent we have

$$\Pr[Y_r = 1] \leq \prod_{r' < r} \left(\kappa + \frac{\delta}{2(r')^2}\right) \leq 2^{1-r} .$$

The last inequality is since $\delta, \kappa \leq 1/3$. We get that the expected number of queries required by the algorithm is at most

$$2 \cdot \sum_{r=1}^{\infty} 2^{-r} \left(H_{\text{explore}} \log(1/\kappa) + (\kappa H_{\text{explore}} + H_{\text{verify}}) \log\left(\frac{2r^2}{\delta}\right)\right) =$$

| MAB task | cite | existing solution | our solution | improvement ratio |
|---|---|---|---|---|
| Dueling Bandits (Condorcet) | [16] | $\left(K^{1+\epsilon}\cdot\sum_{x\neq x^*}\min_{y:\,p_{xy}<0}p_{xy}^{-2}\right)+$ $\sum_{x\neq x^*}\min_{y,p_{xy}<0}p_{xy}^{-2}\log(1/\delta)$ | $\sum_{x\neq x^*}\min\left\{p_{xy}^{-2},\min_{\substack{y'\,:\\p_{xy'}<0}}p_{xy}^{-2}\right\}+$ $\sum_{x\neq x^*}\min_{y,p_{xy}<0}p_{xy}^{-2}\log(1/\delta)$ | $\geq K^{\epsilon}$ for large $\delta$ |
| Linear Bandits | [19] | $\dfrac{d\log(K/\delta)}{\Delta_{\min}^2}$ | $\dfrac{d\log\left(Kd/\Delta_{\min}^2\right)}{\Delta_{\min}^2}+$ $\rho^*(Y^*)\log(1/\delta)$ | up to $d$ for small $\delta$ |
| Unimodal Bandits (line graph) (line graph) | [6] | $\sum_{x\neq x^*}(\Delta_x^{\Gamma})^{-2}+$ $\sum_{x\in\Gamma(x^*)}\Delta_x^{-2}\log(1/\delta)$ | $\sum_{x\neq x^*}(\Delta_x)^{-2}+$ $\sum_{x\in\Gamma(x^*)}\Delta_x^{-2}\log(1/\delta)$ | can be $\Omega(K)$ in typical settings (large $\delta$) |
| Graphical Bandits | [7] | $\dfrac{KD\log(K/\delta)\log^2(K)}{\Delta_{\min}^2}$ | $\dfrac{KD\log^3(K)}{\Delta_{\min}^2}+\dfrac{KD\log(1/\delta)}{\Delta_{\min}^2}$ | $\log^2(K)$ |

Table 1: Comparison between the results obtained by our techniques and the state-of-the-art results in several bandit problem. $K$ represents the total number of arms, $\delta$ the failure probability; in the case of linear bandits, $d$ is the dimension of the space in which the arms lie. The definitions the rest of the problem specific quantities are given in the corresponding sections. The ratio between the solutions, for a typical case is given in the last column.

$$2\cdot\sum_{r=1}^{\infty}2^{-r}\left(H_{\text{explore}}\log(1/\kappa)+(\kappa H_{\text{explore}}+H_{\text{verify}})\log(1/\delta)\right)+$$

$$2\cdot\sum_{r=1}^{\infty}2^{-r}\log(2r^2)\left(\kappa H_{\text{explore}}+H_{\text{verify}}\right)=O\left(H_{\text{explore}}\log(1/\kappa)+(\kappa H_{\text{explore}}+H_{\text{verify}})\log(1/\delta)\right)$$

$\square$

In the following sections we provide algorithms for several bandit problems using the framework of Theorem 3. In Table 1 we provide a comparison between the state-of-the-art results prior to this paper and the results here.

## 4  Application to Dueling Bandits

The *dueling bandit problem*, introduced in [22], arises naturally in domains where feedback is more reliable when given as a pairwise preference (e.g., when it is provided by a human) and specifying real-valued feedback instead would be arbitrary or inefficient. Examples include *ranker evaluation* [14, 23, 12] in information retrieval, ad placement and recommender systems. As with other *preference learning* problems [10], feedback consists of a pairwise preference between a selected pair of arms, instead of scalar reward for a single selected arm, as in the $K$-armed bandit problem.

The formulation of the problem is the following. Given a set of arms $\mathcal{K}$, a query is to a pair $x,y\in\mathcal{K}$ and its output is a r.v. in $\{-1,1\}$ with an expected reward of $P_{ij}$. It is assumed that $P$ is anti-symmetric meaning[5] $P(x,y)=-P(y,x)$ and the $\mu$ values are determined by those of $P$. One common assumption regarding $P$ is the existence of a Condorcet winner, meaning there exist some $x^*\in\mathcal{K}$ for which $P(x^*,y)\geq 0$ for all $y\in\mathcal{K}$. In this case, $x^*$ is defined as the best arm and the reward associated with arm $y$ is typically $P(x^*,y)$. A more general framework can be considered where a Condorcet winner is not assumed to exist. In the absence of a Condorcet winner there is no clear answer as to which arm is the best; several approaches are discussed in [20], [5], and recently in [8, 3], that use some of the notions proposed by social choice theorists, such as the Copeland score or the Borda score to measure the quality of each arm, or game theoretic concepts to determine the best worst-case strategy over arms; we do not elaborate on all of them as they are outside the scope of this paper. In Appendix B.2 we discuss one solution based on the Copeland score, where $\mu(x)$ is defined as the number of arm $y\neq x$ where $P(x,y)>0$.

A general framework capturing both the MAB and DMAB scenarios is that of *partial monitoring games* introduced by [18]. In this framework, when playing an arm $\mathcal{K}$ one obtains a reward $\mu(x)$ yet observes a different function $h(x)$. Some connection between $h$ and $\mu$ is known in advance and based on it, one can design a strategy to discover the best arm or minimize regret. As we do not present results regarding this framework we do not elaborate on it any further, but rather mention that our results, in terms of query complexity, cannot be matched by the existing results there.

## 4.1 Dueling Bandits with the Condorcet Assumption

The Condorcet assumption in the Dueling bandit setting asserts the existence of an arm $x^*$ that beats all other arms. In this section we discuss a solution for finding this arm under the assumption of its existence. Recall that the observable input consists of a set of arms $\mathcal{K}$ of size $K$. There is assumed to exist some matrix $P$ mapping each pair of arms $x, y \in \mathcal{K}$ to a number $p_{xy} \in [-1, 1]$; the matrix $P$ has a zero diagonal, meaning $p_{xx} = 0$ and is anti-symmetric $p_{xy} = -p_{yx}$. A query to the pair $(x, y)$ gives an observation to a random Bernoulli variable with expected value $(1 + p_{xy})/2$ and is considered as an outcome of a match between $x, y$. As we assume the existence of a Condorcet winner, there exists some $x^* \in \mathcal{K}$ with $p_{x^*y} > 0$ for all $y \neq x$.

The Condorcet dueling bandit problem, as stated here and without any additional assumptions was tackled in several papers [20, 26, 16]. The best guarantees to date are given by [16] that provide an asymptotically optimal regret bound for the problem, for the regime of a very large time horizon. This result can be transformed into a best-arm identification algorithm, and the corresponding guarantee is listed in Table 1. Loosely speaking, the result shows that it suffices to query each pair sufficiently many times to separate the corresponding $P_{x,y}$ from 0.5 with constant probability, and additionally only $K$ pairs must be queried sufficiently many times in order to separate the corresponding $P_{x,y}$ from 0.5 with probability $1 - \delta$. We note that other improvements exist that achieve a better constant term (the additive term independent of $\delta$) [25, 24] or an overall improved result via imposing additional assumptions about $P$ such as an induced total order, stochastic triangle inequality etc. [22, 23, 1]. These types of results however fall outside the scope of our paper.

In Appendix B.1 we provide an exploration and verification algorithm for the problem. The exploration algorithm queries all pairs until finding, for each suboptimal arm $x$, an arm $y$ with $p_{xy} < 0$; the exploration algorithm provides as output not only the identity of the optimal arm, but for each sub-optimal arm $x$, the identity of an arm $y(x)$ that (approximately) maximizes $p_{yx}$ meaning it beats $x$ by the largest gap. The verification procedure is now straightforward. Given the above advice the algorithm makes sure that for each allegedly sub-optimal $x$, the arm $y(x)$ indeed beats it meaning $p(yx) > 0$. We obtain the following formal result.

**Theorem 5.** *Algorithm 1, along with the exploration and verification algorithms given in Appendix B.1, finds the Condorcet winner w.p. at least $1 - \delta$ while using an expected amount of at most*

$$\tilde{O}\left(\sum_{y \neq x^*} p_{x^*y}^{-2} + \sum_{x \neq x^*}\sum_{y \neq x} \min\left\{p_{xy}^{-2}, \min_{y', p_{xy'} < 0} p_{xy'}^{-2}\right\}\right) + O\left(\sum_{x \neq x^*} \min_{y, p_{xy} < 0} p_{xy}^{-2} \ln(K/\delta p_{xy}^2)\right)$$

*queries, where $x^*$ is the Condorcet winner.*

## 5 Application to Linear Bandits

The linear bandit problem was originally introduced in [2]. It captures multiple problems where there is linear structure among the available options. Its pure exploration variant (as opposed to the regret setting) was recently discussed in [19]. Recall that in the linear bandit problem the set of arms $\mathcal{K}$ is a subset of $\mathbb{R}^d$. The reward function associated with an arm $x$ is a random variable with expected value $\mu(x) = w^\top x$, for some unknown $w \in \mathbb{R}^d$. For simplicity we assume that all vectors $w$, and those of $\mathcal{K}$ lie inside the Euclidean unit ball, and that the noise is sub-gaussian with variance 1 (hence concentration bounds such as Hoeffding's inequality can be applied).

The results of [19] offer two approaches. The first is a static strategy that guarantees, for failure probability $\kappa$, a query complexity of $\frac{d \log(K/\kappa)}{\Delta_{\min}^2}$ with $x^*$ being the best arm, $\Delta_x = w^\top(x^* - x)$ for $x \neq x^*$ and $\Delta_{\min} = \min_{x \neq x^*} \Delta_x$. The second is adaptive and provides better bounds in a specific case where the majority of the hardship of the problem is in separating the best arm from the second best arm.

The algorithms are based on tools from the area of *Optimal design of experiments* where the high level idea is the following: Consider our set of vectors (arms) $\mathcal{K}$ and an additional set of vecotrs $Y$. We are interested in querying a sequence of $t$ arms from $\mathcal{K}$ that will minimize the maximum variance of the estimation of $w^\top y$, where the maximum is taken over all $y \in Y$. Recall that via the Azuma-Hoeffding inequality, one can show that by querying a set of points $x_1, \ldots, x_t$ and solving the Ordinary Least

Squares (OLS) problem, one obtains an unbiased estimator of $w$ and the corresponding variance to a point $y$ is

$$\rho_{x_1,\ldots,x_t}(y) \triangleq y^\top \left( \sum_{i=1}^t x_i x_i^\top \right)^{-1} y$$

Hence, our formal problem statement is to obtain a sequence $x_1, \ldots, x_t$ that minimizes $\rho_{x_1,\ldots,x_t}(Y)$ defined as $\rho_{x_1,\ldots,x_t}(Y) = \max_{y \in Y} \rho_{x_1,\ldots,x_t}(y)$. Tools from the area of *Optimal design of experiments* (see e.g. [21]) provide ways to obtain such sequences that achieve a multiplicative approximation of $1 + d(d+1)/t$ of the optimal sequence. In particular it is shown that as $t$ tends to infinity, $t$ times the $\rho$ value of the optimal sequence of length $t$ tends to

$$\rho^*(Y) \triangleq \min_p \max_{y \in Y} y^\top \left( \sum_{x \in \mathcal{K}} p_x x x^\top \right)^{-1} y$$

with $p$ restricted to being a distribution over $\mathcal{K}$. We elaborate on these in the extended version of the paper.

[19] propose two and analyze two different choices of the set $Y$. The first is the set $Y = \mathcal{K}$; querying points of $\mathcal{K}$ in order to minimize $\rho_{x_1,\ldots,x_t}(\mathcal{K})$ leads to a best arm identification algorithm with a query complexity of $d \log(K/\kappa)/\Delta_{\min}^2$ for failure probability $\kappa$. We use essentially the same approach for the exploration procedure (given in the extended version), and with the same (asymptotic) query complexity we do not only obtain a candidate best arm $\hat{x}$ but also approximations of the different $\Delta_x$ for all $x \neq x^*$. These are required for the verification procedure.

The second interesting set $Y$ is the set $Y = \left\{ \frac{x^*-x}{\Delta_x} | x \in \mathcal{K}, x \neq x^* \right\}$. Clearly this set is not known to us in advance, but it helps in [19] to define a notion of the 'true' complexity of the problem. Indeed, one cannot discover the best arm without verifying that it is superior to the others, and the set $Y$ provides the best strategy to do so. The authors show that[6]

$$\max_{y \in Y} \|y\|^2 \leq \rho^*(Y) \leq 4d/\Delta_{\min}^2$$

and bring examples where each of the inequalities are tight. Notice that the multiplicative gap between the bounding expressions can be huge (at least linear in the dimension $d$), hence an algorithm with a query complexity depending on $\rho^*(Y)$ as opposed to $d/\Delta_{\min}^2$ can potentially be much better than the above mentioned algorithm. The bound on $\rho^*(Y)$ proves in particular that indeed querying w.r.t. $Y$ is a better strategy than querying w.r.t. $\mathcal{K}$. This immediately translates into a verification procedure. Given the advice from our exploration procedure, we have access to a candidate best arm, and approximate $\Delta$ values. Hence, we construct this set $Y$ and query according to it. We show that given a correct advice, the query complexity for failure probability $\delta$ is at most $O\left(\rho^*(Y^*) \log(K\rho^*(Y^*)/\delta)\right)$. Combining the exploration and verification algorithms, we get the following result.

**Theorem 6.** *Algorithm 1, along with the exploration and verification algorithms described above (we give a the formal version only in the extended version of the paper), finds the best arm w.p. at least $1 - \delta$ while using an expected query complexity of*

$$O\left( \frac{d \log\left(Kd/\Delta_{\min}^2\right)}{\Delta_{\min}^2} + \rho^*(Y^*) \log\left(1/\delta\right) \right)$$

# 6 Application to Unimodal Bandits

The unimodal bandit problem consists of a MAB problem given unimodality information. We focus on a graphical variant defined as follows: There exist some graph $G$ whose vertex set is the set of arm $\mathcal{K}$ and an arbitrary edge set $E$. For every sub-optimal arm $x$ there exist some neighbor $y$ in the graph such that $\mu(x) < \mu(y)$. In other words, the best arm $x^*$ is the unique arm having a superior reward compared to its immediate neighbors. The graphical unimodal bandit problem was introduced by[7] [13].

Due to space constraints we limit the discussion here to a specific type of unimodal bandits in which the underlying graph is line. The motivation here comes from a scenario where the point set $\mathcal{K}$ represents an $\epsilon$-net over the $[0, 1]$ interval and the $\mu$ values come from some unimodal one-dimensional function. We discuss the more general graph scenario only in the extended version of the paper. To review the existing results we introduce some notations. For an arm $x$ let $\Gamma(x)$ denote the set of its neighbors in the graph. For a suboptimal arm $x$ we let $\Delta_x^{\Gamma} = \max_{y \in \Gamma(x)} \mu(y) - \mu(x)$ be the gap between the reward of $x$ and its neighbors and let $\Delta_x = \mu(x^*) - \mu(x)$ be its gap from the best arm $x^*$. We denote by $\Delta_{\min}^{\Gamma}$ the minimal value of $\Delta_x^{\Gamma}$ and $\Delta_{\min}$ be the minimal value of $\Delta_x$. Notice that in reasonable scenarios, for a typical arm $x$ we have $\Delta_x^{\Gamma} \ll \Delta_x$ since many arms are far from being optimal but have a close value to those of their two neighbors.

The state-of-the-art results to date, as far as we are aware, for the problem at hand is by [6], where a method OSUB is proposed achieving an expected query complexity of (up to logarithmic terms independent of $\delta$)[8]

$$O\left( \sum_{x \neq x^*} (\Delta_x^{\Gamma})^{-2} + \sum_{x \in \Gamma(x^*)} \Delta_x^{-2} \log(1/\delta) \right)$$

They show that the summand with the logarithmic dependence over $\delta$ is optimal. In the context of a line graph we provide an algorithm whose exploration is a simple naive application of a best arm identification algorithm that ignores the structure of the problem, e.g. *Exponential Gap-Elimination* by [15]. The verification algorithm requires only the identity of the candidate best arm as advice. It simply applies a best arm identification algorithm over the candidate arm and its neighborhood. The following provides our formal results.

**Theorem 7.** *Algorithm 1, along with the exploration of* Exponential Gap-Elimination *and the verification algorithm of* Exponential Gap-Elimination, *applied to the neighborhood of the candidate best arm, finds the best arm w.p. at least $1 - \delta$ while using an expected query complexity of*

$$O\left( \sum_{x \neq x^*} \Delta_x^{-2} \log\left(K/\Delta_{\min}\right) + \sum_{x \in \Gamma(x^*)} \Delta_x^{-2} \log\left(1/\delta\right) \right)$$

The improvement w.r.t. the results of [6] is in the constant term independent of $\delta$. The replacement of $\Delta_x^{\Gamma}$ with $\Delta_x$ leads to a significant improvement in many reasonable submodular functions. For example, if the arms for an $\epsilon$-net over the $[0, 1]$ interval, and the function is $O(1)$-Lipchitz then $\sum_{x \neq x^*} (\Delta_x^{\Gamma})^{-2} = \Omega(\epsilon^{-3})$ while $\sum_{x \neq x^*} (\Delta_x)^{-2}$ can potentially be $O(\epsilon^{-2})$. Perhaps for this reason, experiments in [6] showed that often, performing UCB on an $\epsilon$-net is superior to other algorithms.

## 7 Conclusions

We presented a general framework for improving the performance of best-arm identification problems, for the regime of high confidence. Our framework is based on the fact that in MAB problems with structure, it is often easier to design an algorithm for verifying a candidate arm is the best one, rather than discovering the identity of the best arm. We demonstrated the effectiveness of our framework by improving the state-of-the-art results in several MAB problems.

## Footnotes

[1]Do to space restrictions we defer the section of Graphical bandits [7] to the extended version.

[2]This objective is naturally extended in the PAC setting where we are interested in an arm that is approximately the best. For simplicity we restrict our focus to the best arm identification problem. We note that our general framework of exploration and verification can be easily expanded to handle the PAC setting as well.

[3]The general form of such algorithms is in fact $H_1 \log(1/\kappa) + H_0$. For simplicity we state our results for the form $H \log(1/\kappa)$; the general statements are an easy modification.

[4]The exact definition of *sufficiently accurate* is given per problem instance.

[5]It is actually common to define the output of $P$ as a number in $[0,1]$ and have $P(x,y)=1-P(y,x)$, but both definitions are equivalent up to a linear shift of $P$.

[6]Under the assumption that all vectors in $\mathcal{K}$ lie in the Euclidean unit sphere

[7]Other variants of the unimodal bandit problem exist, e.g. one where the arms are the scalars in the intervals $[0, 1]$ yet we do not deal with them in this paper, as we focus on pure best arm identification problems and in that scenario the regret setting is more common, and only a PAC algorithm is possible, translating to a $T^{2/3}$ rather than $\sqrt{T}$ regret algorithm

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
