[Supplementary Material · verification_bandit_camera_full.pdf]

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

[9]If the singular value decomposition of $Z$ is $\sum \sigma_i v_i u_i^\top$ then $Z^\dagger = \sum \sigma_i^{-1} u_i v_i^\top$

[10]The result of [6] is in fact tighter in the sense that it takes advantage of the variance of the estimators by using confidence bounds based on KL-divergence. In the case of uniform variance however, the stated results here are accurate. More importantly, the KL-divergence type techniques can be applied here to obtain the same type of guarantees, at the expense of a slightly more technical analysis. For this reason we present the results for the case of uniform variance.

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

# Supplementary Material

## A   Auxiliary Algorithms

In what follows we present algorithms aimed for exploration and verification in several bandit setups. In both cases we would like to bound the expected query complexity of the algorithm, rather than (or in addition to) have a high probability bound on it. The algorithms we present can be directly designed to have a bounded expected query complexity, albeit at the cost of having more technical components in both the algorithm and their proofs. To ease the reading we design these algorithms to have high probability bounds on the query complexity and provide a meta-algorithm below to transform high probability bounds to bounds on the expectation.

Consider the following example to demonstrate the problem dealt here: Consider an algorithm that given some event of probability $1 - \delta$ succeeds and has a query complexity of at most $T$. If this event does not occur the query complexity of the algorithm can be, say, $10^6 T/\delta$. The expected query complexity of the algorithm is much larger than $T$ but given the techniques we provide below it can be reduced to $O(T \log \log(T))$ while keeping the error probability $O(\delta)$. We note that these tricks are rather standard, and for this reason they only appear in the appendix.

### A.1   Some Notation and Observations

In what follows we require some technical notation w.r.t. abstract MAB tasks and algorithms for them. In our setting, a MAB task $\mathcal{T}$ is defined as a problem whose input is an oracle with black box access to the stochastic variables corresponding to the arms (or arm pairs, in the dueling style problems), parametrized by some unknown $\mu \in \mathcal{M}$. In addition to the oracle, the input may also contain some advice vector $\theta^{\text{advice}}$. The possible outputs for a solver of a task is denoted by the set $\Theta$. The output set $\Theta$ in the identification task is the set of arms $\mathcal{K}$. In some cases it will be a Cartesian product of multiple copies of $\mathcal{K}$ and $\mathbb{R}$, as we will require not only the best arm but various other arms, arm-pairs, and estimates of gaps. In verification tasks, the output set $\Theta$ is simply $\{0, 1\}$.

For every characterization $\mu$ and advice vector $\theta^{\text{advice}}$ there exists a subset $\Theta^{\text{correct}} \subseteq \Theta$ of correct outputs. In these, the estimates are sufficiently accurate, and the arm (pairs) identities are as expected. In what follows, our exploration algorithms will be much simpler if designed to have a high probability bound over their query complexity, rather than a bound over their expected query complexity. This is formalized below.

**Definition 8.** *An algorithm $\mathcal{A}$ for a MAB task $\mathcal{T}$ is said to be a* high probability solver *when: given an input characterized by $\mu$ and a failure parameter $\kappa$ it provides the following guarantee: There exists some random event $\mathcal{E}$, whose randomness originates from the randomness of the oracle outputs and possibly the internal randomness of the algorithm, that occurs with probability at least $1 - \kappa$ such that given its occurrence, $\mathcal{A}$ provides a correct output $\theta \in \Theta^{\text{correct}}$. Also, the expected query complexity of $\mathcal{A}$, conditioned on the occurrence of the event $\mathcal{E}$ is at most $T_1 \log(1/\kappa) + T_0$ for some constants $T_1, T_0$ dependent only on the input's characterization $\mu$.*

Our verification algorithms will be much simpler if designed to have a guaranteed bound on their query complexity, only if given a correct advice vector $\theta^{\text{advice}}$. This is formally characterized as follows.

**Definition 9.** *An algorithm $\mathcal{A}$ for a MAB task $\mathcal{T}$ is said to be an* input-dependent high probability solver *when given an input characterized by $\mu$, an advice vector $\theta^{\text{advice}}$ and a failure parameter $\delta$ it provides the following guarantee: There exists some random event $\mathcal{E}$, whose randomness originates from the randomness of the oracle outputs and possibly the internal randomness of the algorithm, that occurs with probability at least $1 - \delta$ such that given its occurrence, $\mathcal{A}$ provides a correct output $\theta \in \Theta^{\text{correct}}$. Also, there exist some set $\Theta^{CI}$ such that if the advice vector $\theta^{\text{advice}} \in \Theta^{CI}$ then the expected query complexity of $\mathcal{A}$, conditioned on the occurrence of the event $\mathcal{E}$ is at most $T_1^V \log(1/\delta) + T_0^V$ for some constants $T_1^V, T_0^V$ dependent only on the input's characterization $\mu$.*

### A.2   Auxiliary Algorithm for Exploration

We now provide a meta-algorithm to transform a *high probability solver* (Definition 8) into an algorithm with a bounded expected query complexity. In Algorithm 2 we invoke the input high

probability solver $\mathcal{A}$ with a query-cap of $T$. This means that if the algorithm does not terminate after $T$ queries it is forcefully stopped.

---

**Algorithm 2** Controlling the Query Complexity

---

**Input:** Oracle access to a high probability solver $\mathcal{A}$ with, failure threshold $\kappa < 1/8$
    **for all** $r = 1, 2, 3, \ldots$ **do**
        Run $\mathcal{A}$ with parameter $\kappa/2r^2$ and query-cap of $2^r$
        If $\mathcal{A}$ terminated before reaching the cap, halt and return its output.
    **end for**

---

**Lemma 10.** *Assume Algorithm 2 is given a high probability solver with guaranteed query complexity of $T_1 \log(1/\kappa) + T_0$, and failure parameter $\kappa < 1/8$. It provides a correct output w.p. at least $1 - \kappa$ and its expected query complexity is at most $O\left(T_1 \log(\log(T_1 + T_0)/\kappa) + T_0\right)$*

*Proof.* The claim regarding the correct output stems from a union bound over all invocations of $\mathcal{A}$. For the query complexity, let $r_0$ be the first value of $r$ for which

$$2^r \geq T_1 \log(r^2/2\kappa) + T_0$$

Notice that for round $r \geq r_0$, given the event in which the expected query complexity of $\mathcal{A}$ is $T_1 \log(r^2/2\kappa) + T_0$, the probability of the algorithm not terminating in round $r$ is, by Markov's inequality, at most

$$\frac{T_1 \log(r^2/2\kappa) + T_0}{2^r} \leq (1 + 2\log(r/r_0)) \cdot \frac{T_1 \log(r_0^2/2\kappa) + T_0}{2^r} \leq$$

$$(1 + 2\log(r/r_0))2^{r_0 - r}$$

Hence, the overall probability of the algorithm not terminating in round $r$ is at most

$$(1 - \kappa)(1 + 2\log(r/r_0))2^{r_0 - r} + \kappa$$

Assuming $r_0 \geq 5$ (otherwise, $T_0, T_1$ are constants and the result is trivial), for $r \geq r_0 + 3$ and $\kappa \leq 1/8$, this probability is at most $1/4$. We get that for $r \geq r_0 + 3$, the probability that the algorithm reached round $r + 1$ is at most $4^{r_0 + 2 - r}$ due to the independence of the algorithm invocations. The expected query complexity is thus bounded by

$$\sum_{r=1}^{r_0 + 2} 2^r + \sum_{r=r_0 + 3}^{\infty} 4^{r_0 + 2 - r}2^r =$$

$$\sum_{r=1}^{r_0 + 2} 2^r + 2^{r_0 + 3} \sum_{r=r_0 + 3}^{\infty} 2^{r_0 + 2 - r} \leq 2 \cdot 2^{r_0 + 3} \leq$$

$$O\left(T_1 \log\left(\log\left(T_1 \log(1/\kappa) + T_0\right)/2\kappa\right) + T_0\right)$$

The last inequality is due to the minimality of $r_0$. $\qquad\square$

### A.3 Auxiliary Algorithm for Verification

In this section we deal with the verification algorithms that are designed to be *input-dependent high probability solver* (Definition 9). Whenever discussing a verification algorithm $\mathcal{A}_V$ it is always a case that we have an exploration algorithm, as discussed above. Given the previous section we formally assume that there exist some exploration algorithm $\mathcal{A}_E$ that does not require any advice and is guaranteed to (1) terminate with an expected query complexity of $T_1^E \log(1/\delta) + T_0^E$ and (2) produce a correct output w.p. at least $1 - \delta$. We may also assume, again by using the result of the above section, that the algorithm $\mathcal{A}_V$, given a correct advice vector $\theta^{\text{advice}} \in \Theta^{\text{CI}}$, has an expected query complexity of $O\left(T_1^V \log(\log(T_1^V + T_0^V)/\delta) + T_0^V\right)$.

The procedure we suggest here is straightforward. Run the verification algorithm $\mathcal{A}_V$ in parallel to $\mathcal{A}_E$. Once one algorithm terminates we can terminate and use its output. This will in the worst case, increase the query complexity by a factor of two.

**Lemma 11.** *Given an advice $\theta^{advice}$, when running $\mathcal{A}_V$ and $\mathcal{A}_E$ in parallel with parameter $\delta$, the output is correct with probability at least $1 - \delta$. If the advice is correct ($\theta^{advice} \in \Theta^{CI}$), the expected query complexity is bounded by*

$$O\left(T_1^V \log(\log(T_1^V + T_0^V)/\delta) + T_0^V\right)$$

*If the advice is incorrect, the expected query complexity is bounded by*

$$2\left(T_1^E \log(1/\delta) + T_0^E\right)$$

# B   Dueling Bandits

In this section we provide exploration and verification algorithms for the Dueling Bandit problem, both for the Condorcet version and the Copeland version.

## B.1   Dueling Bandits with the Condorcet Assumption

### B.1.1   Exploration

Algorithm 3 given below provides the required guarantees for exploration.

---

**Algorithm 3** Exploration in Condorcet Bandits

---

**Input:**  set of arms $\mathcal{K}$, failure probability parameter $\kappa$.
  $Q \leftarrow \{(x,y) \mid x \neq y \in \mathcal{K}\}$, the set of ordered arm pairs.
  **for all** $t = 1 \ldots$ **do**
    query each pair in $\{\{x,y\} \mid (x,y) \in Q \text{ or } (y,x) \in Q\}$ once.
    let $\gamma_t = \sqrt{2\ln(2t^2 K^2/\kappa)/t}$, and let $\ell_{xy}, u_{xy}$ be the lower and upper bounds of $p_{xy}$ according to the confidence interval of radius $\gamma_t$.
    Remove $(x,y)$ from $Q$ if
        •   $\ell_{xy} > 0$
        •   $u_{xy} < 0$ and $2u_{xy} < \ell_{xy}$
        •   $\ell_{xy} < 0$ and $\ell_{xy} > u_{xy'}$ for some $y \neq y'$
    stop when $Q$ is empty
  **end for**
  output $\hat{x}$ as the unique element $x$ for which $\ell_{xy} > 0$ for all $y$. For all $x \neq \hat{x}$, output $y(x) = \arg\min u_{xy}$

---

**Lemma 12.** *It holds with probability at least $1 - \kappa$ that throughout time, for any pair $x, y$ we have $\ell_{xy} \leq p_{xy} \leq u_{xy}$.*

*Proof.* According to Hoeffding's inequality, since the observed r.v.'s for each pair are independent and bounded in $[-1, 1]$ we have that for any specific pair $x, y$ and any time step $t$, the required property holds w.p. at least $1 - \kappa/t^2 K^2$. The claims follows via union bound as there are $K(K-1)/2$ possible pairs. $\square$

**Lemma 13.** *Given the event of Lemma 12, when the algorithm halts we have*

- *The Condorcet winner $x^*$ is the unique arm for which for all $y \neq x^*$ we have $\ell_{x^*y} > 0$*

- *For all arms $x \neq x^*$ let $y(x) = \arg\min u_{xy}$. We have $p_{xy(x)} \leq \frac{1}{2}\min_y p_{xy} < 0$*

*Proof.* For the Condorcet winner $x^*$ we have $p_{x^*y} > 0$ for all $y \neq x^*$ hence the only way a pair $(x^*, y)$ is eliminated from $Q$ is when $\ell_{x^*y} > 0$. Also, for any non-winner $x$ there is some $y$ s.t. $p_{xy} < 0$ hence it cannot be the case that $\ell_{xy} > 0$. This proves the first item. Consider now some $x \neq x^*$.

Define $y'$ as $\arg\min_y p_{xy}$, the arm that beats $x$ by the most. We first show that $y'$ could only have been eliminated according to the elimination rule in Algorithm 3 described in the second bullet. Since

$p_{xy'} < 0$ it must be the case that $\ell_{xy'} < 0$ at all times, hence it could not have been eliminated according to the first bullet. If $y'$ was eliminated according to the third bullet we must have for some other $y$ that

$$p_{xy'} > \ell_{xy'} \geq u_{xy} \geq p_{xy}$$

contradicting the definition of $y'$. It follows that indeed $y'$ is eliminated according to the second bullet and at termination

$$u_{xy'} < 0, \;\; u_{xy'} < \frac{1}{2}\ell_{xy'}$$

Now, according to the definition of $y(x)$ as the minimizer of $u_{xy}$ we have

$$p_{xy(x)} < u_{xy(x)} \leq u_{xy'} < \frac{1}{2}\ell_{xy'} < \frac{1}{2}p_{xy'}$$

as required. $\qquad\square$

After proving the correctness of the algorithm we proceed to analyze its query complexity.

**Lemma 14.** *Given the event of Lemma 12, the algorithm will terminate after at most*

$$\tilde{O}\left(\sum_{y \neq x^*} p_{x^*y}^{-2} + \sum_{x \neq x^*}\sum_{y \neq x} \min\left\{p_{xy}^{-2}, \min_{y': \, p_{xy'} < 0} p_{xy}^{-2}\right\}\right) \ln(1/\kappa)$$

*queries*

*Proof.* Given the definition of $\gamma_t$, for pairs of the form $x^*, y$ it holds that after $O\left(p_{x^*y}^{-2} \log(p_{x^*y}^{-2} K^2/\kappa)\right)$ many queries, $\ell_{x^*y} > 0$, taking care of the left summand in the expression of the Lemma. The same holds for any pair $x, y$ with $p_{xy} > 0$, taking care of the components in the right summand where the dominant part of the min expression is $p_{xy}^{-2}$. For the remaining summands, consider an arm $x \neq x^*$. Let $\Delta_x = |\min_{y', p_{xy'} < 0} p_{xy}|$. Consider a time point $t = O(\log(K/\kappa\Delta_x^2)/\Delta_x^2)$ where we are guaranteed that all of the confidence intervals have a radius of at most $\gamma = \Delta_x/6$. At this time point, for $y' = \arg\min p_{xy}$ we have

$$u_{xy'} < -5\Delta_x/6$$

Now, consider an arm $y$ for which $p_{xy} > -\Delta_x/2$. We have that

$$\ell_{xy} > -\Delta_x/2 - \Delta_x/6 > u_{xy'}$$

hence the pair $xy$ must be eliminated by time $t$. This complies with the minimum expression in the Lemma statement. The remaining type of pairs to discuss are those with $x \neq x^*$ and $y$ with $p_{xy} \leq -\Delta_x/2$. For these we have

$$u_{xy} < -\Delta_x/3, \;\; \ell_{xy} > -2\Delta_x/3 > 2u_{xy}$$

and the pair $xy$ is indeed eliminated by time $t$. The claim follows. $\qquad\square$

### B.1.2 Verification

**Lemma 15.** *Assume that $\hat{x} = x^*$ and for all $x \neq x^*$, $p_{xy(x)} \leq \frac{1}{2}\min_y p_{xy}$. Then w.p. at least $1 - \delta$ the algorithm halts after*

$$O\left(\sum_{x \neq x^*} \min_{y, p_{xy} < 0} p_{xy}^{-2} \ln(K/\delta p_{xy}^2)\right)$$

*many queries, and outputs 'success'*

*Proof.* We start with the observation that w.p. at least $1 - \delta$ all of the confidence intervals contain the relevant $p_{xy}$ throughout time (this is true due to Hoeffding's inequality and a simple union bound argument). For $x \neq x^*$ let $\Delta_x = \max_y p_{yx}$ and let $t_x = c\Delta_x^{-2}\ln(K/\delta\Delta_x^2)$ for some sufficiently large constant $c$. Due to the guarantee for $y(x)$, the guarantee about the correctness of the confidence

---

**Algorithm 4** Condorcet Dueling Bandits Verification

---

**Input:** set of arms $\mathcal{K}$, failure probability parameter $\delta$, candidate Condorcet winner $\hat{x}$, candidate adversary $y(x)$ for each $x \neq \hat{x}$ in $\mathcal{K}$.

$Q \leftarrow \{x \mid x \neq \hat{x} \in \mathcal{K}\}$

**for all** $t = 1 \ldots$ **do**

    For every $x \in Q$, query the pair $(x, y(x))$ once

    let $\gamma_t = \sqrt{2ln\left(2t^2K^2/\delta\right)/t}$, and let $\ell_{xy}, u_{xy}$ be the lower and upper bounds of $p_{xy}$ according to the confidence interval of radius $\gamma_t$.

    Remove $x$ from $Q$ if $u_{xy(x)} < 0$

    if $\ell_{xy(x)} > 0$ for some $x$, terminate with an answer 'fail'

    if $Q$ becomes empty, terminate with an answer of 'success'

**end for**

---

intervals, and their definition we have that after $t_x$ rounds it must be the case that $u_{xy(x)} < 0$. It follows that the total number of queries is

$$\sum_{x \neq x^*} t_x = O\left(\sum_{x \neq x^*} \min_{y, p_{xy} < 0} p_{xy}^{-2} \ln(K/\delta p_{xy}^2)\right)$$

as required. To prove the correctness of the output, notice that since $p_{xy(x)} < 0$ it cannot be the case that $\ell_{xy(x)} > 0$ for any $x$ hence the algorithm will only terminate when all arms are removed from $Q$ hence its output will be 'success'. $\qquad\square$

**Lemma 16.** *Assume that $\hat{x} \neq x^*$. Then w.p. at least $1 - \delta$, when the algorithm terminates it outputs 'fail'*

*Proof.* Recall that w.p. at least $1 - \delta$ all of the confidence intervals contain the relevant $p_{xy}$ throughout time. We assume this is indeed the case and show the output must be 'fail'. Since $\hat{x} \neq x^*$ it must be the case that some $x \neq \hat{x}$ is the Condorcet winner. For this arm, $p_{xy(x)} > 0$. Hence, throughout time we must have $u_{xy(x)} > 0$ and if the algorithm terminates it cannot be because of $Q$ being empty, but only due to $\ell_{x'y(x')} > 0$ for some $x'$, meaning the output will be 'fail'. $\qquad\square$

Theorem 5 is now an immediate corollary of the above results, combined with those of Appendix A.

### B.2 Copeland Dueling Bandits

In this section we analyze a natural extension of the Condorcet winner denoted as the Copeland winner. This approach was suggested in several papers, e.g. [20, 5, 24]. The Copeland score of an arm $x$ is the number of arm it beats, i.e. $\sum_{y \neq x} \mathbf{1}[p_{xy} > 0]$. Notice that if the Condorcet winner exists then the arm with the maximal Copeland score is unique and is indeed the Condorcet winner. We assume throughout for simplicity that for all $x \neq y$, $p_{xy} \neq 0$ meaning there are no ties. Extending our results to a setting where ties can occur is purely technical and will cause complex guarantees, hence we do not discuss the issue any further.

The state-of-the-art identification procedure can be achieved via an easy modification of the regret-based results of [24]. We note mention that a very recent result by [17] improves the regret based Copeland dueling bandit paper of [24], yet it is not clear how to transform it into a high probability best arm identification algorithm. A loose description of the query complexity of the CCB algorithm given in [24] is $K^2 + K(s+1)\log(1/\delta)$ for failure probability $\delta$, where $s$ is the number of losses suffered by a Copeland winner. They provide convincing arguments as to why in practice, the value of $s$ is constant despite the theoretical possibility of it scaling as $K$. The exact query complexity guarantee is (up to logarithmic terms)

$$O\left(\sum_{x \neq y}\left(|p_{yx}| + \max\left\{0, \max_{y' \neq x}^{s+1} p_{y'x}\right\}\right)^{-2} + \left(\sum_{x \in C, y \notin C} p_{xy}^{-2} + \sum_{x \notin C}\frac{s+1}{\left(\max_{y \neq x}^{s+1} p_{yx}\right)^2}\right)\log(1/\delta)\right)$$

Here, $C$ is the set of Copeland winners, and we use $\max^k$ to denote an operator returning the $k$'th largest value from a set. We note that in the same paper an additional algorithm (SCB) is presented with an incomparable query complexity. The term that is related to $\delta$ is larger, but there is no quadratic dependence in $K$, assuming the quantities of $1/p_{xy}^2$ are large, specifically $p_{xy} \gg 1/\sqrt{K}$ for all $x, y$. For the purpose of brevity we do not elaborate on this result further.

Our methods can be used to improve upon the above result via rather simple algorithms. In what follows we present an exploration and verification algorithm. The exploration algorithm queries arm pairs uniformly and eliminates them once either it is clear whether $p_{xy} > 0$ or vice versa, or both $x$ and $y$ can be excluded from being a Copeland winner regardless of whether $x$ beats $y$ or vice versa. The advice that is eventually produced consists of (1) the identity of a Copeland winner $\hat{x}$ (2) the identity of $K - 1 - s$ arms $\hat{y}_1, \ldots, \hat{y}_{K-1-s}$ that $\hat{x}$ beats by the largest gap, and (3) for every arm $x \neq \hat{x}$, the identity of $s$ arms $y_1^x, \ldots, y_s^x$ that beat $x$ by the largest gaps. Also, the provided Copeland winner is the one for which the gaps in (2) are the largest in the sense that it requires the least queries to verify that it beats the respective $\hat{y}$'s with high confidence. We show (Lemma 18) that the query complexity required to achieve this advice correctly w.p. at least $1 - \kappa$ is $O\left(\sum_{x \neq y} p_{xy}^{-2} \log\left(K/\kappa p_{xy}^2\right)\right)$. Given the above advice our verification algorithm verifies that indeed $\hat{x}$ is a Copeland winner. It queries the pairs $x, y_i^x$ uniformly until it discovers, with high confidence, that $p_{y_i^x x} > 0$ for all $x \neq \hat{x}, i$ and queries the pairs $\hat{x}, \hat{y}_i$ until realizing $p_{\hat{x}\hat{y}_i} > 0$ for all $i$. If this is indeed the case then we have with high confidence that all arms other than $\hat{x}$ are beaten by at least $s$ arms, and $\hat{x}$ is beaten by at most $s$ arms, hence $\hat{x}$ is indeed a Copeland winner w.h.p. If any of the above inequalities do not hold, the algorithm outputs 'fail'. The query complexity, for failure probability $\delta$, given a good advice vector is easily shown to be $H_{\text{cplnd}}^{\text{v}} \log(1/\delta)$ for

$$
H_{\text{cplnd}}^{\text{v}} = \tilde{O}\left( \min_{x^* \in C} \min_{y_1, \ldots, y_{K-1-s} \mid p_{x^* y_i} > 0} \sum_i p_{x^* y_i}^{-2} + \sum_{x \neq x^*} \min_{y_1, \ldots, y_s \mid p_{xy} < 0} \sum_i p_{xy_i}^{-2} \right)
$$

Theorem 17 describes the guarantees given by the described exploration and verification algorithms, when combined with the techniques of Algorithm 1. It is an easy task to verify that in the regime of small $\delta$, the expression is strictly smaller than the guarantee obtained by [24]. The exact ratio depends on the structure of the different $p'_{xy}s$.

**Theorem 17.** *Algorithm 1, along with the exploration and verification algorithms given below, finds a Copeland winner w.p. at least $1 - \delta$ while using an expected amount of at most*

$$
O\left( \sum_{x \neq y} p_{xy}^{-2} \log\left(K/p_{xy}^2\right) + H_{\text{cplnd}}^{\text{v}} \log(1/\delta) \right)
$$

*queries.*

### B.2.1 Exploration

**Lemma 18.** *w.p. at least $1 - \kappa$ we have*

1. *If there is more than one Copeland winner, let $s$ denote the number of losses suffered by the Copeland winners. Otherwise, let $s$ be such that $s + 1$ is the smallest amount of losses suffered by a non-Copeland winner*

2. *Upon termination, $s_t = s$.*

3. *(minimality of $x^*$) If there is more than one Copeland winner then $x^*$ is a Copeland winner with*

$$
\sum_{y \mid p_{x^* y} > 0} p_{x^* y}^{-2} \leq 4 \min_{x \in C} \sum_{y \mid p_{xy} > 0} p_{xy}^{-2}
$$

   *with $C$ being the set of Copeland winners.*

4. *(minimality of $y_i^*$)*

$$
\sum_{i=1}^{K-1-s} p_{x^* y_i^*}^{-2} \leq 4 \min_{y_1, \ldots, y_{K-1-s} \mid p_{xy_i} > 0} \sum_i p_{x^* y_i}^{-2}
$$

---

**Algorithm 5** Copeland Bandits Exploration

---

**Input:** set of arms $\mathcal{K}$, failure probability parameter $\kappa$.

  $Q \leftarrow \{(x,y) \mid x \neq y\}$

  **for all** $t = 1 \dots$ **do**

    query each pair in $\{\{x,y\} \mid (x,y) \in Q$ or $(y,x) \in Q\}$ once

    let $\gamma_t = \sqrt{2ln\left(2t^2 K^2/\kappa\right)/t}$, and let $\ell_{xy}, u_{xy}$ be the lower and upper bounds of $p_{xy}$ according to the confidence interval of radius $\gamma_t$.

    For arm $x$, let $L_\ell(x) = |\{y \mid u_{xy} < 0\}|$, $L_u(x) = |\{y \mid \ell_{xy} < 0\}|$ be the lower and upper bounds on the number of losses of $x$.

    Set $s_t = \min_{x \in \mathcal{K}} L_u(x)$

    Let $B = \{x \mid L_\ell(x) > s_t\}$ be the set of arms that can be excluded from being a Copeland winner.

    Remove $(x,y)$ from $Q$ if

      •   $\ell_{xy} > 0$ and $2\ell_{xy} > u_{xy}$

      •   $u_{xy} < 0$ and $2u_{xy} < \ell_{xy}$

      •   $x \in B$ and $\ell_{xy} > 0$

      •   $x \in B$, $\ell_{xy} < 0$ and $\ell_{xy} > \min_{y_1,\dots,y_{s_t}} \max_{i \in [s_t]} u_{xy_i}$

    If $Q$ is empty, terminate and:

      •   Output $x^*$ as the arm not in $B$, minimizing $\min_{y_1,\dots,y_{K-1-s_t} \mid \ell_{x^*y_i} > 0} \sum_{i=1}^{K-1-s_t} \ell_{x^*y_i}^{-2}$

      •   Output $y_1^*, \dots, y_{K-1-s_t}^*$, the minimizers of the above expression

      •   For every $x \neq x^*$, output $\{y_i(x)\}_{i=1}^{s_t}$, the minimizers of $\min_{y_1,\dots,y_{s_t}} \max_{i \in [s_t]} u_{xy_i}$

  **end for**

---

5. *(minimality of $y_i(x)$) For any $x \neq x^*$ we have*

$$\sum_{i=1}^{s} p_{xy_i(x)}^{-2} \leq 4 \min_{y_1,\dots,y_{s_t} \mid p_{xy} < 0} \max_{i \in [s_t]} p_{xy_i}^{-2}$$

6. *The query complexity of the algorithm is at most*

$$O\left(\sum_{x \neq y} p_{xy}^{-2} \ln(K/\kappa p_{xy}^2)\right)$$

*Proof.* We prove the lemma based on the event that all confidence intervals contain $p_{xy}$ for all $x,y$ pairs, throughout time. It is an easy exercise to see that this event happens w.p. at least $1 - \kappa$.

Given the event of the confidence intervals being accurate it is clear that $L_u(x), L_\ell(x)$ are indeed upper and lower bounds on the losses of an arm $x$. As a result, any arm $x \in B$ cannot be a Copeland winner since being in $B$ interprets into having some arm $x' \neq x$ with $L_u(x') < L_\ell(x)$.

To prove item 2, it suffices to show that $s_t$ is not too large upon termination. For $s_t$ to be too large we must have that for all $x \in C$, $L_u(x)$ is strictly larger than the true number of losses of $x$. This implies that there exist some $(x,y)$ pair with $x \in C$ such that $u_{xy} > 0$ and $\ell_{xy} < 0$. This pair could not have been eliminated according to the first and second bullets describing the elimination. Since $x \in C$ cannot be in the set $B$, the pair $x,y$ cannot be eliminated according to the third and fourth bullets either. It follows that while $s_t > s$, the set $Q$ cannot be empty and the algorithm will not terminate.

We proceed to prove items 3 and 4. A conclusion of the above discussion is that upon termination, we have for all pairs $x,y$ where $x$ is a Copeland winner that $\ell_{xy} > 2u_{xy} > 0$ or $u_{xy} < 0$. In case there is more than one Copeland winner we have that for $x^*$,

$$\sum_{y \mid p_{x^*y} > 0} p_{x^*y}^{-2} \leq \sum_{y \mid p_{x^*y} > 0} \ell_{x^*y}^{-2} =$$

$$\min_{x \in C} \sum_{y \mid p_{xy} > 0} \ell_{xy}^{-2} \leq \min_{x \in C} \sum_{y \mid p_{xy} > 0} 4u_{xy}^{-2} \leq$$

$$\min_{x \in C} \sum_{y \;|p_{xy}>0} 4 p_{xy}^{-2}$$

Also,

$$\sum_{i=1}^{K-1-s} p_{x^* y_i^*}^{-2} \le \sum_{i=1}^{K-1-s} \ell_{x^* y_i^*}^{-2} =$$

$$\min_{y_1,\dots,y_{K-1-s} \;|p_{x^* y_i}>0} \sum_i \ell_{x^* y_i}^{-2} \le$$

$$4 \min_{y_1,\dots,y_{K-1-s} \;|p_{x^* y_i}>0} \sum_i u_{x^* y_i}^{-2} \le$$

$$4 \min_{y_1,\dots,y_{K-1-s} \;|p_{x^* y_i}>0} \sum_i p_{x^* y_i}^{-2}$$

We proceed to analyze Item 5. Notice first that due to $s_t = s$, for all arms $x \ne x^*$ we have upon termination knowledge of at least $s$ arms $y$ that beat $x$, meaning $u_{xy} < 0$. If follows that

$$\sum_{i=1}^{s} p_{xy_i(x)}^{-2} \le \sum_{i=1}^{s} u_{xy_i(x)}^{-2} = \min_{y_1,\dots,y_s \;|\; u_{xy_i}<0} \sum_{i=1}^{s} u_{xy_i}^{-2} \le$$

$$\min_{y_1,\dots,y_s \;|\; u_{xy_i}<0} 4 \sum_{i=1}^{s} \ell_{xy_i}^{-2} \le \min_{y_1,\dots,y_s \;|\; u_{xy_i}<0} 4 \sum_{i=1}^{s} p_{xy_i}^{-2}$$

Next, consider an arm $y$ for which $p_{xy} < 0$ but upon termination it holds that $u_{xy} > 0$. It must be the case that for some arm $y'_x(y)$,

$$0 > p_{xy} > \ell_{xy} > u_{xy'_x(y)} > p_{xy'_x(y)}$$

and we have that

$$\min_{y_1,\dots,y_s \;|\; u_{xy_i}<0} 4 \sum_{i=1}^{s} p_{xy_i}^{-2} \le \min_{y_1,\dots,y_s \;|\; p_{xy_i}<0} 4 \sum_{i=1}^{s} p_{xy_i}^{-2}$$

thus proving Item 5.

It remains to analyze the query complexity. Notice that once $0 < 0.5 u_{xy} < \ell_{xy}$ or $0 > 0.5\ell_{xy} > u_{xy}$, both the (ordered) pairs $(x,y),(y,x)$ are eliminated from $Q$ and hence the unordered pair $x,y$ is not queried again. Since $p_{xy}$ lies within the confidence region, and its radius at time $t$ scales as $\sqrt{\log(Kt/\kappa)/t}$, we get that $x,y$ can be queried no more than $O(p_{xy}^{-2} \log(K/\kappa p_{xy}^2))$ times before being eliminated from $Q$. $\qquad\square$

### B.2.2 Verification

---
**Algorithm 6** Copeland Bandits Verification

---
**Input:** set of arms $\mathcal{K}$, failure probability parameter $\delta$, candidate winner $\hat{x}$ and $K-s$ different arms $\hat{y}_1,\dots,\hat{y}_{K-s}$, for every $x \ne \hat{x}$, $s$ arms $y_1(x),\dots,y_s(x)$.
    $Q \leftarrow \{(\hat{x},\hat{y}_i) \mid i \in [K-s]\} \cup \{(x,y_i(x)) \mid x \ne \hat{x}, \; i \in [s]\}$
    **for all** $t = 1\dots$ **do**
        query each pair in $\{\{x,y\} \mid (x,y) \in Q \text{ or } (y,x) \in Q\}$ once
        let $\gamma_t = \sqrt{2ln\left(2t^2 K^2/\delta\right)/t}$, and let $\ell_{xy}, u_{xy}$ be the lower and upper bounds of $p_{xy}$ according to the confidence interval of radius $\gamma_t$.
        If $u_{\hat{x}\hat{y}_i} < 0$ for some $i \in [K-s]$, terminate and output 'fail'
        If $\ell_{xy_i(x)} > 0$ for some $x \ne \hat{x}, i \in [s]$, terminate and output 'fail'
        Remove from $Q$ all $(x,y)$ for which the corresponding confidence region does not contain zero.
        If $Q$ is empty, terminate and output 'success'
    **end for**

---

**Lemma 19.** *If there is more than one Copeland winner, let $s$ denote the number of losses suffered by the Copeland winners. Otherwise, let $s$ be such that $s+1$ is the smallest amount of losses suffered by a non-Copeland winner. Assume the advice is proper, meaning that:*

*1. $\hat{x}$ is the Copeland winner minimizing, up to a multiplicative term of 4, the expression*

$$\sum_{y \mid p_{xy} > 0} p_{xy}^{-2}$$

*2.*

$$\sum_{i=1}^{K-1-s} p_{\hat{x}\hat{y}_i}^{-2} \le 4 \min_{y_1,\ldots,y_{K-1-s} \mid p_{\hat{x}y_i} > 0} \sum_i p_{\hat{x}y_i}^{-2}$$

*3. For any $x \ne \hat{x}$ we have*

$$\sum_{i=1}^{s} p_{xy(x)}^{-2} \le 4 \min_{y_1,\ldots,y_{s_t} \mid p_{xy} < 0} \max_{i \in [s_t]} p_{xy(x)}^{-2}$$

*Then, w.p. at least $1 - \delta$ Algorithm 6 outputs 'success' and has a query complexity of at most*

$$O\left( \min_{x^* \in C} \min_{y_1,\ldots,y_{K-1-s} \mid p_{x^*y_i} > 0} \sum_i p_{x^*y_i}^{-2} \log(Kp_{x^*y_i}^{-2}/\delta) + \sum_{x \ne x^*} \sum_{y_1,\ldots,y_s \mid p_{xy} < 0} p_{xy_i}^{-2} \log(Kp_{xy_i}^{-2}/\delta) \right)$$

*Proof.* We notice that w.p. at least $1 - \delta$, for all pairs $x, y$ and throughout time $p_{xy}$ lies inside the corresponding confidence interval. Given this event, and the fact that the radius of the confidence intervals scale as $\sqrt{\log(t/\delta)/t}$ we get that after

$$O\left( \min_{x^* \in C} \min_{y_1,\ldots,y_{K-1-s} \mid p_{x^*y_i} > 0} \sum_i p_{x^*y_i}^{-2} \log(Kp_{x^*y_i}^{-2}/\delta) + \sum_{x \ne x^*} \sum_{y_1,\ldots,y_s \mid p_{xy} < 0} p_{xy_i}^{-2} \log(Kp_{xy_i}^{-2}/\delta) \right)$$

many queries, no confidence interval contains zero and the algorithm terminates. This proves the query complexity. The correctness follows from the fact that $p_{xy}$ is always contained in the confidence intervals and that the advice is proper, as detailed in the claim. $\square$

**Lemma 20.** *Given an advice with $\hat{x}$ that is not a Copeland winner, the probability of Algorithm 6 giving an output of 'success' is at most $\delta$.*

*Proof.* Assume that for all $\hat{y}_1,\ldots,\hat{y}_{K-1-s}$ it holds that $p_{\hat{x}\hat{y}_i} > 0$. This means that $\hat{x}$ suffers at most $s$ losses. Assume further that for all $x \ne \hat{x}$ and all $i \in [s]$, $p_{xy_i(x)} < 0$. This means that all $x \ne \hat{x}$ suffer at least $s$ losses. It follows that these assumptions lead to $\hat{x}$ being a Copeland winner. Since the claim is for the opposite case, we must have $p_{\hat{x}\hat{y}_i} < 0$ for some $i$ or $p_{xy_i(x)} > 0$ for some $i$.

Hence, if the confidence intervals are always correct, the corresponding pair will never be eliminated from $Q$ and the only possible output for the algorithm is 'fail'. It follows that in order for the algorithm to output 'success' some confidence interval must be wrong and this event happens w.p. at most $\delta$, as required. $\square$

Theorem 17 is now an immediate corollary of the above results, combined with those of Appendix A.

## C Linear Bandits

### C.1 Auxilary Tools

In this section we analyze algorithms that query the unknown vector $w$ at various points and obtain a confidence region for it. We make use of the following lemma providing a high probability confidence region given a set of queried points.

**Lemma 21.** *Let $y$ be an arbitrary vector in the unit sphere and let $z_1,\ldots,z_t$ be an arbitrary sequence of vectors in the unit sphere. Let $r_1,\ldots,r_t$ be noisy outcomes of queries to $\{w^\top z_i\}_i$ where the noise has mean zero, absolute value of at most 1 almost surely, and is independent in each $i$. Let $A = \sum z_i z_i^\top$, $b = \sum z_i r_i$ and $\hat{w} = A^{-1}b$. It holds for any $\alpha > 0$ that*

$$\Pr\left[ \left| (w - \hat{w})^\top y \right| > \sqrt{\alpha y^\top A^{-1} y} \right] \le 2\exp\left(-\alpha/2\right)$$

*Proof.* Denote by $Z$ the matrix whose $i$'th column is $z_i$. Denote by $\epsilon_i$ the noise in query $i$ meaning $r_i = w^\top z_i + \epsilon_i$ and let $\epsilon$ be the vector of length $t$ with the different $\epsilon_i$ values. According to the definition of $\hat{w}$ we see that

$$y^\top (w - \hat{w}) = y^\top Z^\dagger \epsilon = \sum (y^\top Z^\dagger)_i \epsilon_i$$

for $Z^\dagger$ being the pseudo-inverse[9] matrix of $Z$. It follows that the error associated with $y$ is a martingale sum and according to the Azuma-Hoeffding inequality it can be bounded by

$$\Pr\left[\left|(w - \hat{w})^\top y\right| > \sqrt{\alpha}\|y^\top Z^\dagger\|\right] \le 2\exp\left(-\alpha/2\right)$$

Since $Z^\dagger (Z^\dagger)^\top = A^{-1}$ the claim follows □

In both the exploration and verification algorithms, we use a fixed strategy for querying points. In particular we query a sequence of points $z_1, \ldots, z_t$ that aim to minimize

$$\rho(z_1, \ldots, z_t) = \max_{y \in Y} y^\top \left(\sum_i z_i z_i^\top\right)^{-1} y$$

For some fixed set $Y$. We follow a common approach in the area of *optimal design of experiments* and choose the $z$ vectors greedily. At each time step we choose the point that minimizes the above expression. In [19], appendix C, a review of this method is given stating that for

$$\rho^*(Y) = \min_p \max_{y \in Y} y^\top \left(\sum_{x \in \mathcal{K}} p_x x x^\top\right)^{-1} y$$

where $p$ is restricted to be a distribution over $\mathcal{K}$, it holds that for any $t$,

$$\rho^*(Y) \le t \cdot \rho(z_1, \ldots, z_t) \le \rho^*(Y) \cdot (1 + d(d+1)/t) \tag{1}$$

Additionally, in the case where $Y = \mathcal{K}$, it is also known that

$$\rho^*(\mathcal{K}) \le d \tag{2}$$

In the following section we will aim to find a sequence corresponding to a set $Y$ while having access to a different set $Y' = \{\alpha(y) \cdot y \mid y \in Y\}$, where the $\alpha(y)$'s are arbitrary scalars in $[1/c, c]$ for some constant $c > 1$. The following observation provides a guarantee for a sequence w.r.t. $Y$ given its guarantee w.r.t. $Y'$.

**Lemma 22.** *Let $Y \subseteq \mathbb{R}^d$, let $Y' = \{\alpha(y) \cdot y \mid y \in Y\}$, where the $\alpha(y)$'s are arbitrary scalars in $[1/c, c]$, with $c > 1$, and let $x_1, \ldots, x_t \in \mathcal{K}$. Assume that*

$$c\rho^*(Y') \ge t \max_{y \in Y'} y^\top \left(\sum_{i=1}^t x_i x_i^\top\right)^{-1} y$$

*then we have that*

$$c^5 \rho^*(Y) \ge t \max_{y \in Y} y^\top \left(\sum_{i=1}^t x_i x_i^\top\right)^{-1} y$$

*Proof.* Let

$$p_Y^* = \arg \min_{p \in \Delta_{|\mathcal{K}|}} \max_{y \in Y} y^\top \left(\sum_{x \in \mathcal{K}} p(x) x x^\top\right)^{-1} y$$

be the distribution over the possible set of points achieving $\rho^*(Y)$.

$$\rho^*(Y) = \max_{y \in Y} y^\top \left(\sum_{x \in \mathcal{K}} p_Y^*(x) x x^\top\right)^{-1} y \ge$$

$$\frac{1}{c^2} \max_{y \in Y'} y^\top \left( \sum_{x \in \mathcal{K}} p_Y^*(x) x x^\top \right)^{-1} y \geq \frac{1}{c^2} \rho^*(Y')$$

Hence, we get that

$$t \max_{y \in Y} y^\top \left( \sum_{i=1}^t x_i x_i^\top \right)^{-1} y \leq$$

$$c^2 t \max_{y \in Y'} y^\top \left( \sum_{i=1}^t x_i x_i^\top \right)^{-1} y \leq c^3 \rho^*(Y') \leq c^5 \rho^*(Y)$$

$\square$

## C.2 Exploration

---
**Algorithm 7** Linear Bandits Exploration

---
**Input:** set of arms $\mathcal{K}$, failure probability parameter $\kappa$.

  $A_0 \leftarrow I$

  **for all** $t = 1, 2, 3, \ldots$ **do**

    Pick a point $z_t \in \mathcal{K}$ minimizing $\max_{x \in \mathcal{K}} x^\top (A_{t-1} + zz^\top) x$

    Update $A_t = A_{t-1} + z_t z_t^\top$, $b = \sum_{i=1}^t r_i z_i$, $\hat{w} = A_t^{-1} b$

    For $x \in \mathcal{K}$ let $\gamma_x = \sqrt{x^\top A_t^{-1} x \ln(K 4 t^2 / \kappa)/2}$

    If there exist some $\hat{x} \in \mathcal{K}$ such that

$$3(\gamma_x + \gamma_{\hat{x}}) < \hat{w}^\top (\hat{x} - x)$$

    for all $x \neq \hat{x}$, output $\hat{x}$ as the best arm and $\hat{\Delta}_x = \hat{w}^\top (\hat{x} - x)$ for all $x \neq \hat{x}$.

  **end for**

---

The following lemma is immediate given a union bound and Lemma 21.

**Lemma 23.** *w.p. at least $1 - \kappa$ it holds that for all $t > 0$ and all $x \in \mathcal{K}$ that*

$$\left| (w - \hat{w})^\top x \right| < \gamma_x$$

**Lemma 24.** *Let $x^*$ be the best arm, $\Delta_x = w^\top(x^* - x)$ and $\Delta_{\min} = \min_x \Delta_x$. Given the event of Lemma 23 the algorithm terminates after at most $O\left(\max\left\{d^2, d\ln(Kd/\Delta_{\min}\kappa)/\Delta_{\min}^2\right\}\right)$ queries. Also, $\hat{x}$ is the best arm and $0.5\Delta_x \leq \hat{\Delta}_x \leq 1.5\Delta_x$ for all suboptimal $x$.*

*Proof.* We begin with correctness of the algorithm: Since it always holds that

$$\left| (w - \hat{w})^\top x \right| < \gamma_x$$

for all $x$, upon termination we have that for any $x \neq \hat{x}$,

$$w^\top (\hat{x} - x) > \hat{w}^\top (\hat{x} - x) - \gamma_x - \gamma_{\hat{x}} > 2\gamma_x + 2\gamma_{\hat{x}} > 0$$

meaning that $\hat{x}$ is indeed the optimal arm. It follows that $\Delta_x > 2\gamma_x + 2\gamma_{\hat{x}}$. Since

$$\left| (\hat{w} - w)^\top (\hat{x} - x) \right| < \gamma_x + \gamma_{\hat{x}} < \Delta_x/2$$

the claim regarding $\hat{\Delta}_x$ holds.

We now turn to bounding the query complexity. For $t \geq d(d+1)$ we have for all $x \in \mathcal{K}$ by Equations (1) and (2) that

$$\gamma_x \leq \sqrt{\rho^*(\mathcal{K}) \ln(K 4 t^2 / \kappa)/t} \leq \sqrt{d \ln(K 4 t^2 / \kappa)/t}$$

It follows that for $t \geq 64 d \ln(K 4 t^2 / \kappa)/\Delta_{\min}^2$ we have

$$\gamma_x < \Delta_{\min}/8$$

for all $x$. For such $t$,

$$\hat{w}^\top (x^* - x) \geq w^\top (x^* - x) - \gamma_x - \gamma_{x^*} >$$
$$3\Delta_{\min}/4 \geq 3(\gamma_x + \gamma_{x^*})$$

and the algorithm will terminate after $O\left(d\ln(Kd/\Delta_{\min}^2\kappa)/\Delta_{\min}^2\right)$ queries as required

$\square$

## C.3 Verification

---

**Algorithm 8** Linear Bandits Verification

---

**Input:** set of arms $\mathcal{K}$, failure probability parameter $\delta$, candidate winner $\hat{x}$, for any $x \neq \hat{x}$ a parameter $\hat{\Delta}_x > 0$.

$A_0 \leftarrow I$

$Y \leftarrow \{ \frac{\hat{x}-x}{\hat{\Delta}_x} \mid x \in \mathcal{K},\ x \neq \hat{x} \}$

**for all** $t = 1, 2, 3, \ldots$ **do**

    Pick a point $z_t \in \mathcal{K}$ minimizing $\max_{y \in Y} y^\top (A_{t-1} + zz^\top) y$

    Update $A_t = A_{t-1} + z_t z_t^\top$, $b = \sum_{i=1}^{t} r_i z_i$, $\hat{w} = A_t^{-1} b$

    For $x \in \mathcal{K}$ let

$$\gamma_x = \sqrt{(\hat{x} - x)^\top A_t^{-1} (\hat{x} - x) \ln(K4t^2/\delta)/2}$$

    If

$$\gamma_x < \hat{w}^\top (\hat{x} - x)$$

    for all $x \neq \hat{x}$, output 'success'

    if there exist $x \neq \hat{x}$ for which

$$\gamma_x < \hat{w}^\top (x - \hat{x})$$

    output 'fail'

**end for**

---

As mentioned in the main paper, the query complexity of the algorithm will be expressed in terms of $\rho^*(Y^*)$ defined as

$$\rho^*(Y^*) = \min_p \max_{x \in \mathcal{K}, x \neq x^*} \frac{(x^* - x)^\top \left( \sum_{x \in \mathcal{K}} p_x x x^\top \right) (x^* - x)}{\Delta_x^2}$$

where the minimum is taken over distributions over $\mathcal{K}$

The following lemma is immediate given a union bound and Lemma 21.

**Lemma 25.** *w.p. at least* $1 - \delta$ *it holds that for all* $t > 0$ *and all* $x \in \mathcal{K}$ *that*

$$\left| (w - \hat{w})^\top (\hat{x} - x) \right| < \gamma_x$$

**Lemma 26.** *If* $\hat{x}$ *is the best arm and* $\Delta_x/2 \leq \hat{\Delta}_x \leq 1.5\Delta_x$ *for all* $x \neq \hat{x}$ *then given the event of Lemma 25 the algorithm will output 'success' and will terminate after at most* $t = O\left( \max\{ d^2, \rho^*(Y^*) \ln(K\rho^*(Y^*)t/\delta) \} \right)$ *queries.*

*Proof.* In order for the algorithm to output 'fail' it must be that

$$\hat{w}^\top (x - \hat{x}) > \gamma_x > w^\top (x - \hat{x}) + \gamma_x$$

which is assumed not to happen, hence the algorithm must output 'success'.

To analyze the query complexity, For $t \geq d(d+1)$ we have for all $x \in \mathcal{K}$ by Equation 1 that

$$\gamma_x < \sqrt{\rho^*(Y) \Delta_x^2 \ln(K4t^2/\delta)/t}$$

hence if $t \geq \sqrt{\rho^*(Y) \ln(K4t^2/\delta)}$ we get that $\gamma_x < \Delta_x$, meaning that for all $x \neq \hat{x}$,

$$\hat{w}^\top (\hat{x} - x) > w^\top (\hat{x} - x) - \gamma_x > 0$$

and the algorithm will terminate. The claim follows from noticing that since $\Delta_x/2 \leq \hat{\Delta}_x \leq 1.5\Delta_x$ it must be the case that $\rho^*(Y) \leq 32\rho^*(Y^*)$ (Lemma 22). $\qquad\square$

**Lemma 27.** *If* $\hat{x}$ *is not the best arm then w.p. at least* $1 - \delta$ *the algorithm will output 'fail'*

*Proof.* We prove the lemma conditioned on the occurrence of the event of Lemma 25. Since $\hat{x}$ is suboptimal there must be some $x \in \mathcal{K}$ for which $w^\top(\hat{x} - x) < 0$. For the algorithm to output 'success' it must be the case that

$$\hat{w}^\top(\hat{x} - x) > \gamma_x > w^\top(\hat{x} - x) + \gamma_x \ .$$

But, according to our assumption this can never happen, hence the algorithm must output 'fail' when it terminates. □

Theorem 6 is now an immediate corollary of the above results, combined with those of Appendix A.

## D Unimodal Bandits

### D.1 Unimodal Bandits for General Graphs

In this section we present the existing results, as well as our own for the unimodal bandit problem with general graphs. To review the existing results we introduce some notations. We denote by $d$ the maximum degree of the graph. For an arm $x$ let $\Gamma(x)$ be the set of arms $y$ in the immediate neighborhood of $x$ in the graph. For a suboptimal arm $x$ we let $\Delta_x^\Gamma = \max_{y \in \Gamma(x)} \mu(y) - \mu(x)$ be the gap between the reward of $x$ and its neighbors and let $\Delta_x = \mu(x^*) - \mu(x)$ be its gap from the best arm $x^*$. We denote by $\Delta_{\min}^\Gamma$ the minimal value of $\Delta_x^\Gamma$ and $\Delta_{\min}$ be the minimal value of $\Delta_x$. Notice that $\Delta_{\min}^\Gamma \leq \Delta_{\min}$ and that the ratio between the two is potentially unbounded. Furthermore, it is very often the case that for most $x$, $\Delta_x^\Gamma \ll \Delta_x$ as $x$ may be a clear bad choice compared to the optimal arm but still have a very close value to those of its immediate neighbors. Consider a subset of the edges forming a spanning tree $T$ over the graph. We say that $T$ is traversable if it preserves the unimodality property, meaning that every $x \neq x^*$ there exist some neighbor in $T$ with a superior reward. Denote by $D(T)$ the diameter, i.e. the longest shortest path between a pair of vertices, in $T$, and denote by $D$ the maximum value of $D(T)$ with the maximum taken over all possible traversable trees $T$.

The method GLSE in [13], though aimed for the regret setting can be proved to achieve, for failure probability $\kappa$ an expected query complexity of $O\left(Dd(\Delta_{\min}^\Gamma)^{-2}\log(1/\kappa) + D\log(D)\right)$ for the special case where the graph is a tree, in which case $D$ is simply the diameter of the tree. In [6], a method OSUB is proposed achieving an expected query complexity of (up to logarithmic terms independent of $\kappa$)[10]

$$O\left(\sum_{x \neq x^*}(\Delta_x^\Gamma)^{-2} + \sum_{x \in \Gamma(x^*)}\Delta_x^{-2}\log(1/\kappa)\right)$$

for general graphs. The latter result has a better dependency over $\kappa$, and they in fact prove that it is asymptotically optimal when $\kappa$ tends to 0. However, when compared to GLSE while discussing trees, in some cases the size of the graph could be as large as $K = d^{\Omega(D)}$ in which case the linear dependence over $\mathcal{K}$ can lead to inferior results compared to those of [13].

Our methods lead to two algorithms, differing only in the exploration strategy, each improving a different result of those mentioned above. In the first setting we use the idea of [13] and jump from one vertex to the next, while always increasing the reward of the visited arm. The algorithm is detailed in Appendix D, and achieves an expected query complexity of $O\left(Dd\log\left(Dd/\kappa\Delta_{\min}^\Gamma\right)/(\Delta_{\min}^\Gamma)^2\right)$. The second exploration algorithm is a simple naive application of a best arm identification algorithm that ignores the structure of the problem, e.g. *Exponential Gap-Elimination* [15] that achieves an expected query complexity of

$$O\left(\sum_{x \neq x^*}\Delta_x^{-2}\log\left(\log(1/\Delta_x)/\kappa\right)\right) \leq O\left(K\Delta_{\min}^{-2}\log\left(\log(1/\Delta_{\min})/\kappa\right)\right)$$

In both cases, the verification algorithm requires only the identity of the candidate best arm as advice. It simply applies a best arm identification algorithm over the candidate arm and its neighborhood. Its expected query complexity, given a correct advice is $O\left(\sum_{x\in\Gamma(x^*)}\Delta_x^{-2}\log\left(\log(1/\Delta_x)/\delta\right)\right)$. The following provides our formal results.

**Theorem 28.** *Algorithm 1, along with the exploration algorithm detailed in Appendix D.2 and the verification algorithm of* Exponential Gap-Elimination, *applied to the neighborhood of the candidate best arm, finds the best arm w.p. at least $1-\delta$ while using an expected query complexity of*

$$O\left(Dd\log\left(Dd/\Delta_{\min}^{\Gamma}\right)/(\Delta_{\min}^{\Gamma})^2 + \sum_{x\in\Gamma(x^*)}\Delta_x^{-2}\log\left(1/\delta\right)\right)$$

*queries. When applied with the exploration algorithm of* Exponential Gap-Elimination, *it achieves an expected query complexity of*

$$O\left(\sum_{x\neq x^*}\Delta_x^{-2}\log\left(K/\Delta_{\min}\right) + \sum_{x\in\Gamma(x^*)}\Delta_x^{-2}\log\left(1/\delta\right)\right)$$

Notice that the first result strictly improves that of [13], while being applicable to both trees and general graphs, and the second result improves, in the terms independent of $\delta$, the result of [6], as $\Delta_{\min} \geq \Delta_{\min}^{\Gamma}$.

## D.2 Exploration Algorithm

We proceed to provide an exploration algorithm for the graphical unimodal bandit problem. We begin by presenting a sub-procedure that visits a single node in the graph in Algorithm 9.

---

**Algorithm 9** Node Visit in Graphical Unimodal Bandits

---

**Input:** set of arms $\Gamma$, additional arm $x$ and confidence parameter $\kappa$.
  $Q \leftarrow \Gamma \cup \{x\}$
  **for all** $t = 1, 2, 3, \ldots$ **do**
    Query each arm in $Q$ once
    Let $\hat{\mu}(y)$ be the empirical reward of arm $y \in Q$.
    Set
$$\gamma = \sqrt{2\ln\left(2(|\Gamma|+1)t^2/\kappa\right)/t}$$

    Eliminate all arms $y \in Q$ for which $\hat{\mu}(x) - \hat{\mu}(y) > \gamma$
    If there exists an $y \in Q$ for which $\hat{\mu}(y) - \hat{\mu}(x) > \gamma$, output $y$
    If $Q$ is empty, output 'x'
  **end for**

---

To provide the analysis of the algorithm we introduce some notations. For an arm $x$ and set $\Gamma$, let $\Delta = \max\{\max_{y\in\Gamma}\mu(y) - \mu(x), \min_{y\in\Gamma}\mu(x) - \mu(y)\}$.

**Lemma 29.** *With probability at least $1-\kappa$ it holds that Algorithm 9 (1) terminates within*

$$O\left(|\Gamma|\log\left(|\Gamma|/\kappa\Delta^2\right)/\Delta^2\right)$$

*queries, and (2) if $x$ has the maximal value among $\Gamma \cup \{x\}$ then the output is x, otherwise the output is some $y$ with $\mu(y) > \mu(x)$.*

*Proof.* Recall that our model assumption dictates that the random variables of $\mu(y)$ are in the region $[0, 1]$, hence according to Hoeffding's inequality we have that for any arm $y$ and any time $t$, w.p. at least $1 - \kappa/2(|\Gamma|+1)t^2$, $|\hat{\mu}(y) - \mu(y)| < \gamma/2$. It follows via union bound that for all arms and all time steps $|\hat{\mu}(y) - \mu(y)| < \gamma/2$. The claim immediately follows. $\square$

Our exploration algorithm, given in Algorithm 10, consists of applying the above procedure in order to traverse the graph while increasing the $\mu$ value until reaching $x^*$.

---

**Algorithm 10** Graphical Unimodal Bandits Exploration

---

**Input:** set of arms $\mathcal{K}$, confidence parameter $\kappa$.

    Pick an arbitrary arm $x_1$.

    **for all** $r = 1, 2, 3, \ldots$ **do**

        If $r = 1$, set $\Gamma = \Gamma(x_1)$, else set $\Gamma = \Gamma(x_r) \setminus \{x_{r-1}\}$.

        Invoke Algorithm 9 with input $x_r, \Gamma, \kappa' = \kappa/2r^2$

        If the output is $x_r$, halt and output $x_r$ as the best arm. Otherwise, for an output $y$ proceed to the next iteration with $x_{r+1} = y$.

    **end for**

---

The following is immediate given the definitions of $D$ and $\Delta_{\min}^{\Gamma}$ given in the previous section and the Lemma 29.

**Lemma 30.** *With probability at least $1 - \kappa$ it holds that Algorithm 10 (1) terminates within*

$$O\left(Dd \log\left(Dd/\kappa\Delta_{\min}^{\Gamma}\right)/(\Delta_{\min}^{\Gamma})^2\right)$$

*queries, and (2) outputs the best arm $x^*$.*

Theorem 28 is now an immediate corollary of the above results, combined with those of Appendix A.

## E  Application to Graphical Bandits

The *Graphical Bandits* problem is a variant of the dueling bandit problem presented in [7]. As in the dueling bandit scenario, in each round the user plays a pair of arms $(x, y)$. The difference is an additional restriction where not all pairs are valid but only a subset of the pairs denoted by $E$. In their paper, the authors assume that each arm has an associated reward $\mu(x) \in [0, 1]$ and that the outcome of a query to $(x, y)$ is a random variable in $[-1, 1]$ with an expected value of $\mu(x) - \mu(y)$. While other variations may be worth considering, we restrict ourselves to the same assumptions here for simplicity.

In [7], the complexity of the problem is tied to the diameter of the graph, meaning the largest distance in edges in the graph $G(\mathcal{K}, E)$ between a pair of arms. Denoting the diameter as $D$, they provide a best arm identification problem that fails w.p. at most $\kappa$ and requires an expected query complexity of

$$\frac{KD\log(K/\kappa)\log^2(K)}{\Delta_{\min}^2}$$

with $\Delta_{\min}$ being the gap between the best and second best arm. We use the exact same algorithm for the exploration, as the verification process requires only the identity of a candidate best arm. Given this candidate $\hat{x}$, our verification procedure is defined as follows. It first computes a shortest path (in edges) tree originating from $\hat{x}$, denoted by $T$. Next, it performs an elimination tournament where at each round the edges leading to surviving arms are queried. That is, once an arm $x$ is known w.h.p. to be beaten by $\hat{x}$, $x$ is removed from the set of surviving arms and any edges in $T$ that do not lead to any other surviving arm, will not be queried again. The formal algorithm and its analysis are given in Appendix E.1. To present its guarantee we introduce a few notations. Let $x^*$ be the best arm and let $\Delta_x = \mu(x^*) - \mu(x)$ be the gap for a suboptimal arm. Let $T$ be the shortest path tree originating from $x^*$ produced by the BFS (breadth first search) algorithm given and let $d(x)$ be the distance from $x^*$ to $x$ in the graph. Let $E(T)$ be the set of edges in $T$ and for an edge $e$ in $E(T)$ let $h_e = \max_x d(x)/\Delta_x^2$ where the maximum is taken over arms $x$ whose path towards them in $T$ contains the edge $e$. Our verification algorithm in Appendix E.1, given a correct advice and tuned for failure probability $\delta$, achieves an expected query complexity of

$$O\left(\sum_{e \in E(T)} h_e \log\left(Kh_e/\delta\right)\right) \leq O\left(\frac{KD\ln(K/\delta)}{\Delta_{\min}^2}\right)$$

As a corollary, given the techniques of Theorem 3 we get:

**Theorem 31.** *Algorithm [1], along with the exploration algorithm of [7] and the verification algorithms given in Appendix E.1, finds the best arm w.p. at least $1 - \delta$ while using an expected amount of at most*

$$O\left( \frac{KD\log\left(KD/\Delta_{\min}\right)\log^2(K)}{\Delta_{\min}^2} + \sum_{e\in E(T)} h_e \log\left(Kh_e/\delta\right) \right) \leq$$

$$O\left( \frac{KD\left(\log\left(KD/\Delta_{\min}\right)\log^2(K) + \log(1/\delta)\right)}{\Delta_{\min}^2} \right)$$

*queries.*

## E.1 Verification in Graphical Bandits

Algorithm [11] is our verification algorithm. For $x \in \mathcal{K}$, denote by $\tilde{\Delta}_x = \mu(\hat{x}) - \mu(x)$ the gap between rewards of the candidate arm $\hat{x}$ and that of $x$. In the algorithm we keep a confidence interval around the empirical estimation $\hat{\Delta}_x$ of $\tilde{\Delta}_x$, of radius $\gamma_x$. We use an adaptation of Proposition 1 in [7] to our terminology to prove the required a high probability confidence bound.

---

**Algorithm 11** Graphical Bandits Verification

---

**Input:** set of arms $\mathcal{K}$, a graph structure $G = (\mathcal{K}, E)$, and a candidate winner $\hat{x}$.
    compute the shortest path tree (in edges) originating from $\hat{x}$
    $Q \leftarrow \mathcal{K} \setminus \{\hat{x}\}$
    For every edge $e$ in the tree, keep $n_e \leftarrow 0, \hat{\mu}_e \leftarrow 0$.
    **for all** $t = 1, 2, 3, \ldots$ **do**
        For arm $x$ denote by $\pi(x)$ the path from $\hat{x}$ to $x$ in the tree.
        Let $E' = \cup_{x\in Q}\pi(x)$
        query each $e \in E'$ once
        Let $\hat{\mu}_e$ be the empirical estimation of the expected value returned by a query to an edge $e$. Set

$$\hat{\Delta}_x = \sum_{e\in\pi(x)} \hat{\mu}_e \,,$$

$$\gamma_x = \sqrt{2|\pi(x)|\ln(4Kt^2/\delta)/t}$$

        If $\hat{\Delta}_x + \gamma_x < 0$ for any $x$, output 'fail'
        Remove all $x$ meeting $\hat{\Delta}_x - \gamma_x > 0$ from $Q$
        If $Q$ is empty, output 'success'
    **end for**

---

**Lemma 32.** *For any $x \in \mathcal{K}$ and any time step $t$ we have w.p. at least $1 - \delta/2Kt^2$ that*

$$\left| \tilde{\Delta}_x - \hat{\Delta}_x \right| < \gamma_x$$

*Proof.* Due to the definition of $\tilde{\Delta}_x$ we have that for our estimator $\hat{\Delta}_x$ it holds that

$$\hat{\Delta} = \tilde{\Delta} + \frac{1}{t}\sum_{i=1}^{|\pi(x)|t} \epsilon_i$$

with $\epsilon_i$ being independent zero mean noise terms in $[-1, 1]$. By Hoeffding's inequality we have that for any $\gamma > 0$,

$$\Pr\left[ \frac{1}{|\pi(x)|}\left| \tilde{\Delta}_x - \hat{\Delta}_x \right| < \gamma \right] \leq 2\exp(-\gamma^2|\pi(x)|t/2)$$

In particular, for $\gamma_x$ we get

$$\Pr\left[ \left| \tilde{\Delta}_x - \hat{\Delta}_x \right| < \gamma_x \right] \leq 2\exp(-\gamma_x^2 t/2|\pi(x)|) = 2\exp(-\ln(4Kt^2/\delta)) = \frac{\delta}{2Kt^2}$$

$\square$

As a corollary, via a simple union bound argument we obtain

**Lemma 33.** *With probability at least* $1 - \delta$ *it holds that for all* $t$ *and all* $x$

$$\left| \tilde{\Delta}_x - \hat{\Delta}_x \right| < \gamma_x$$

**Lemma 34.** *If* $\hat{x}$ *is not the optimal arm then w.p. at least* $1 - \delta$ *the algorithm will output 'fail'*

*Proof.* Since $\hat{x}$ is not the optimal arm there is some arm $x$ s.t. $\tilde{\Delta}_x < 0$. If the algorithm provided an output of 'success' then $x$ must have been eliminated from $Q$. At that point we must have had

$$\hat{\Delta}_x > \gamma_x > \tilde{\Delta}_x + \gamma_x$$

and this event can occur w.p. of at most $\delta$ □

**Lemma 35.** *Assume that* $\hat{x}$ *is the best arm. Let* $e$ *be an edge in* $T$, *the shortest path tree originating from* $x^*$, *computed by our algorithm. Let* $\mathcal{K}(e)$ *be the set of arms for which* $e \in \pi(x)$, *and let* $h_e = \max_{x \in \mathcal{K}(e)} |\pi(x)| \Delta_x^{-2}$. *Then w.p. at least* $1 - \delta$ *the algorithm will output 'success' and use at most* $O\left( \sum_{e \in E(T)} h_e \ln(K h_e / \delta) \right)$ *queries.*

*Proof.* We analyze the algorithm given the occurrence of the event of Lemma 33, that happens w.p. at least $1 - \delta$. We begin with the proof of correctness. If the algorithm provided an output of 'fail' then for some $x$ must have had

$$\hat{\Delta}_x < -\gamma_x < \tilde{\Delta}_x - \gamma_x$$

where the last inequality holds since $\hat{x}$ is the best arm hence $\tilde{\Delta}_x = \Delta_x > 0$. This is a contradiction.

We continue to analyze the query complexity. Consider an arbitrary sub-optimal arm $x$ and let $t$ be such that

$$t \geq 8 \Delta_x^{-2} |\pi(x)| \ln(4K t^2 / \delta)$$

We have that $\gamma_x \leq \Delta_x / 2$ and

$$\hat{\Delta}_x > \Delta_x - \gamma_x \geq \gamma_x$$

meaning that $x$ is eliminated from $Q$ at that time. The claim regarding the query complexity immediately follows. □