[Reviews · NeurIPS 2016]

Reviewer 1

Summary

The paper considers the best arm identification (BAI) problem, with fixed confidence, in several structured multi-armed bandit settings. Intuitively, in these settings, verification (of an arm being the best) is an easier problem than the identification of the best arm. Starting from this observation, a new "explore-verify" framework is introduced in the paper. Based on tools from previous on BAI, explore-verify algorithms are proposed for BAI in linear, dueling, unimodal, and graphical bandits settings. The algorithms are analyzed and compared to the state-of-the-art results.

Qualitative Assessment

The paper proposes a novel explore-verify framework for the well-studied best-arm identification problem and applies it to several structured bandit settings previously studied in the literature. The paper is nicely written and the proposed explore-verify algorithms are clearly presented. A merit of the paper is that it shows how this framework can be adapted to a wide range of setups, by adapting it to four types of structure in multi-armed bandits. The proposed algorithms start from some of the techniques employed by previous algorithms in the literature, and design specific Exploration and Verification rules for each problem. The analysis is sound, presented in detail, and allows to obtain an improvement in the four settings. On the other hand, no experiment is included in the paper. It would have been nice to see for at least one of the settings how the performance improves in practice. For instance, an example where there is a large enough \delta such that the Explore-Verify Condorcet Bandit algorithm behaves better than previous algorithms. Typos: Abstract: Mutli-armed --> Multi-armed Lines 197,374,422: it's --> its

Confidence in this Review

2-Confident (read it all; understood it all reasonably well)


Reviewer 2

Summary

The authors consider the MAB problem with i.i.d. rewards, but where the reward function has a structure. They propose a general algorithmic idea for this problem, and apply it to several structured bandit problems known earlier in the litterature. Their algorithmic idea is roughly as follows: exploration is split in two phases, in the first phase one looks for a candidate arm (an arm which is possibly optimal), and in the second phase one determines whether or not the candidate arm is indeed optimal. If the outcome is positive the algorithm has identified the optimal arm, otherwise one returns to the first phase. The authors argue that, while the second phase is the one responsible for most of the regret (for small confidence parameter delta), it is usually easier to design for some structured problems (for instance the unimodal structure where one must sample neighbours of the optimal arm).

Qualitative Assessment

The paper is overall interesting and clearly written. While the idea of verification is indeed simple, it seems strong enough to bring improvement to known results for several popular types of structures. It should be made clear that what the authors propose is more of an algorithmic idea which needs to be tailored for each structured bandit problem. For instance, in order to design the verification phase efficiently, one needs to know precisely what is the structure at hand. It would also be good if the paper included numerical experiments in order to show how the proposed algorithms behave in practice, in particular how sensitive they can be to the exploration parameter \kappa (given \delta fixed). Small remark: the Table 1 which presents the relative improvement with respect to previous solutions is a bit unclear at places, due to the fact that it is not always clear which term dominates. It could be better if two columns are added (one for delta close to 1 and another one for delta close to 0). For instance, as far as I understand, for Unimodal Bandits, the improvement can only be \Omega(K) in settings where K is much larger than \log(1/delta). Indeed, the algorithm of [6] is asymtotically optimal, so that if the O( \log(1/delta) ) term dominates, there is no possible improvement. Also, I might have missed something, but in the linear bandit case, isnt the lower bound O( d log(1/delta) / \Delta^2 ) ? Therefore I do not understand how one can obtain a d-fold improvement over the result [19]. This should be clarified.

Confidence in this Review

1-Less confident (might not have understood significant parts)


Reviewer 3

Summary

This paper proposes a verification-based framework for solving a range of bandit problems, including condorcet dueling bandits, copeland dueling bandits, linear bandits, unimodal bandits, and graphical bandits. The setting considered is PAC-style guarantees for pure exploration, rather than online regret minimization. The authors show an iterative approach whereby a verification algorithm is used to verify a proposed solution by an exploration algorithm. Since the verification algorithm has cheaper sample complexity, one can construct a meta-algorithm that iteratively calls the two algorithms in order to achieve rigorous PAC-style sample complexity guarantees. These guarantees are compared against existing results, and the authors demonstrate improvements in some regimes. In general, I think the technical results are interesting and mostly sound, although I can't help but feel that the presentation could be significantly improved. Some of the results seem a bit incomplete, although I think a more complete theoretical account is straightforward. My larger concern is the significance of the results in the paper. Part of this could be resolved by greater clarity in the writing. More details are covered in the sections below. As it stands, my review is borderline. I can see this paper as suitable for NIPS or a bit below the accept threshold, depending on the author response.

Qualitative Assessment

*** Significance Part 1 *** I wonder if the authors have looked into the details of the proofs of some of the previous work. One common trick that people do in these types of theoretical analyses is to simplify the results because one cares only about the simpler bounds that merges together a more complicated summation of terms into a single term. For example, the result for Linear Bandits shows an improvement that could be non-existent with a more refined analysis of the result in [19]. This would be somewhat analogous to what the authors noted in footnote 3. *** Significance Part 2 *** The authors argue that these results are meaningful improvements upon previous work. However, it would be easier to accept this argument if the authors can construct explicit cases that satisfy the results. For instance, I'm particularly interested in the linear case. Numerical experiments could also help and/or complement the theoretical results. *** Clarity *** The paper feels a bit incomplete in its writing. For instance, Copeland dueling bandits isn't substantially discussed at all in the main paper. And related to the above point on significance, the reader feels a bit disappointed by a lack of substantial discussion on the implications of the theoretical results. Furthermore, the paper doesn't actually formally define any problem settings. I realize this is very difficult due to the generality of the framework being proposed, but this lack of specificity is somehow dissatisfying. *** Proofs *** I'm curious about the equations after Line 135. It seems like there is an extra kappa*H_explore term -- can the authors explain this? Perhaps related to the above question, the authors never explicitly state a sample complexity of the verification oracle when the findbestarm subroutine fails. For instance, Lemma 14 doesn't have a sample complexity guarantee. I think this should be a straightforward fix, but the theoretical results are incomplete without them.

Confidence in this Review

3-Expert (read the paper in detail, know the area, quite certain of my opinion)


Reviewer 4

Summary

The authors provide a framework for the best arm identification problem, in the fixed confidence setting. The idea is to find an empirical best arm with a small number of observations (and with a lower probability of success than the targeted confidence) and then to verify if indeed this arm is optimal, this time with targeted probability. This process is repeated until a positive verification. The authors argue that the verification problem allows to match (or even outperform in some case) the sample complexity of existing algorithms on various best arm identification problems (dueling bandits/linear bandits/...).

Qualitative Assessment

The proposed approach is quite interesting and is, at my knowledge, innovative. Even, if the idea itself is good, they are some flaw in the results and in the presentation of the paper. 1/ In think that the comparisons between sample complexities of the new methods and the state of the art could be unfair. Results from this paper are given in expectation (with respect to probability of the 'find then verify iterative process') whereas state of the art sample complexity results are given in high probability and therefore are stronger. 2/ Even if an high probability result is maybe hard to achieve, some experiments may help to convince of the effectiveness of the approach. With some indicators like the mean time needed to find the best arm, the number of times when the new method finds the best arm before the state of the art (for me, the more useful indicator with regard to the discrepancy between the theoretical results, the expectation may involve a lot of variance in practice) 3/ The applications of the generic framework to the Dueling Bandits/Linear Bandits/... are at the same time too much detailed and not enough. Right now, half of the paper describe these applications while being not self contained and too much dependant from the appendix. Maybe choosing one and detailing it could clarify the second part of the paper. You could then notice the reader that the same process can be applied to other setting. With the current presentation, a major part of the appendix could have their place in the paper.

Confidence in this Review

2-Confident (read it all; understood it all reasonably well)


Reviewer 5

Summary

The paper proposes a meta-algorithm for finding the best arm in structured MAB problems. The algorithm consists of two components. One is "Find_Best_Arm" that outputs a candidate best arm with some additional information, which could be an existing algorithm for best arm identification. The other is "Verify_Best_Arm" that uses the output of Find_Best_Arm and determines whether the candidate arm is the best arm or not. It provides some analysis to show why the interleaving of the two components may lead to better query complexity. The meta-algorithm is applied to Dueling bandit, Linear Bandit, Unimodal Bandit etc, which improves the regret upper bounds of existing works.

Qualitative Assessment

Regards to the second bullet and its proof in Lemma 11, what does \min_y p_{ x y(x) } means? As y(x) = \argmin u_{xy}, the \min_y operator causes confusion. For its proof, last line in page 12 in the supplementary states "p_{ x y(x) } \geq l_{ x y(x) } \geq 2 u_{ x y(x) } \geq 2 u_{ x y' } \geq 2 p_{ x y'}", why does "2 u_{ x y(x) } \geq 2 u_{ x y' } " holds? It seems to contradicts the definition of y(x). For Lemma 13, it assumes p_{ x y(x) } \leq 2 min_y p_{xy}. When does the assumption holds? Which parts of the proof in Lemma 13 use the assumption? This paper would be better if some simulation results would be provided to support the analysis. Minors: In the supplementary line 419: "let \Delta_x = \max_y p_{xy}" should be "let \min_y p_{xy}" "let t_x = c \Delta_x^{-2} \log( K / delta Delta_x )" should be "let t_x = c \Delta_x^{-2} \log( K / delta Delta_x^2 )" "we have that after t_x rounds it must be the case that u_xy \le 0" ---> u_{x y(x)} \le 0 line 510: "\Delta_x \leq \hat{ \Delta_x } \leq 1.5 \Delta_x for all suboptimal x ..." should be "0.5 \Delta_x \leq \hat{ \Delta_x } \leq 1.5 \Delta_x"

Confidence in this Review

2-Confident (read it all; understood it all reasonably well)


Reviewer 6

Summary

This paper introduces a method called "Explore-Verify" to solve pure exploration bandit problem. The author uses this framework to improve algorithms in various of pure-exploration bandit problems.

Qualitative Assessment

The idea of "Explore-Verify Frame" is very good. But the improvement ratio of all settings in this paper needs some special settings like "\delta is small" or "\delta is large". So a question is whether there is some other solutions that do not need to use this framework given these special cases. Some questions about the proofs: 1) In the proof of Lemma 8: why using Marcov Inequality can have the probability is at most 2^{r_0-r}? I think using Marcov Inequality the probability upper bound is (T_1\log(r^2/2k)+T_0)/2^r. But the numerator is not upper bounded by 2^r_0. 2) In the proof of Lemma 12, whythe dominate part is p_{xy}^{-2} when p_{xy} > 0? Why we can ignore the p_{xy}^{-2} in the logarithm term?

Confidence in this Review

2-Confident (read it all; understood it all reasonably well)